# Giant honeybees (Apis dorsata) trade off defensiveness against periodic mass flight activity

**Gerald Kastberger** ᴼ*, **Martin Ebner, Thomas Hötzl**

Institute of Biology, University of Graz, Graz Austria

* gerald.kastberger@uni-graz.at

**Data Availability Statement:** All relevant data are within the manuscript and, at least exemplified, in the Supporting Information Movie files.

## Abstract

The giant honeybee *Apis dorsata* (Fabricius, 1793) is an evolutionarily ancient species that builds its nests in the open. The nest consists of a single honeycomb covered with the bee curtain which are several layers of worker bees that remain almost motionless with their heads up and abdomens down on the nest surface, except for the mouth area, the hub between inner- and outer-nest activities. A colony may change this semi-quiescence several times a day, depending on its reproductive state and ambient temperature, to enter the state of mass flight activity (MFA), in which nest organisation is restructured and defense ability is likely to be suppressed (predicted by the *mass-flight-suspend-defensiveness* hypothesis). For this study, three episode of MFA ($mfa_{1-3}$) of a selected experimental nest were analysed in a case study with sequences of >60 000 images at 50 Hz, each comprise a short pre-MFA session, the MFA and the post-MFA phase of further 10 min. To test colony defensiveness under normative conditions, a dummy wasp was cyclically presented with a standardised motion programme ($P_d$) with intervening sessions without such a presentation ($nP_d$). Motion activity at five selected surveillance zones ($sz_{1-5}$) on the nest were analysed. In contrast to $mfa_{1,2}$, in $mfa_3$ the experimental regime started with the cyclic presentation of the dummy wasp only after the MFA had subsided. As a result, the MFA intensity in $mfa_3$ was significantly lower than in $mfa_{1-2}$, suggesting that a colony is able to perceive external threats during the MFA. Characteristic ripples appear in the motion profiles, which can be interpreted as a start signal for the transition to MFA. Because they are strongest in the mouth zone and shift to higher frequencies on their way to the nest periphery, it can be concluded that MFA starts earlier in the mouth zone than in the peripheral zones, also suggesting that the mouth zone is a control centre for the scheduling of MFA. In $P_d$ phases of pre- and postMFA, the histogram-based motion spectra are biphasic, suggesting two cohorts in the process, one remaining at quiescence and the other involved in shimmering. Under MFA, $nP_d$ and $P_d$ spectra were typically Gaussian, suggesting that the nest mates with a uniform workload shifted to higher motion activity. At the end of the MFA, the spectra shift back to the lower motion activities and the $P_d$ spectra form a biphasic again. This happens a few minutes earlier in the peripheral zones than in the mouth zone. Using time profiles of the skewness of the $P_d$ motion spectra, the *mass-flight-suspend-defensiveness* hypothesis is confirmed, whereby the inhibition of defense ability was found to increase progressively during the

**Funding:** Funding: This work was supported by the Austrian Science Foundation (FWF Project P 20515-B16) URL: https://www.fwf.ac.at/en/. The funders had no role in the study design, data collection and analysis, decision to publish or preparation of the manuscript. The authors gratefully acknowledge the financial support for the publication costs by the University of Graz.

**Competing interests:** The authors have declared that no competing interests exist.

MFA. These sawtooth-like time profiles of skewness during MFA show that defense capability is recovered again quite quickly at the end of MFA. Finally, with the help of the $P_d$ motion spectra, clear indications can be obtained that the giant honeybees engage in a decision in the sense of a tradeoff between MFA and collective defensiveness, especially in the regions in the periphery to the mouth zone.

## Introduction

Giant honeybees (clade *Megapis*) are evolutionarily among the oldest honeybee species, with an age of more than 10 million years. To date, two species are recognized in this subgenus, the first being *Apis dorsata* (Fabricius, 1793), which is distributed in the lowland southern regions of Asia in several ecotypes (or even potential species) from Pakistan in the west to the Philippines in the east and from Sri Lanka in the south to the foothills of the Himalayas in the north [1–4]. The second species, *A. laboriosa* (Smith), 1871 [5–8] occurs from the Himalayan valley floors preferably up to an altitude of 3,200 m or even higher, with its range extending from Pakistan to northern Vietnam. Both species are extremely migratory [3, 9–13] and regularly switch from singularized nests hidden in forests to the reproductive status, where they form colony aggregations at traditional roost sites [7, 9, 14]. They have the ability to exploit niches in food sources that are spurned even by other flower-visiting species and, on the other hand, can anticipate their migratory cycles to take advantage of the seasonal supply of local agriculture [12, 13, 15], which distinguishes them as a synanthropic species.

Giant honeybees build their nests in the open, fixing the top of the nest at a substrate, such as a rocky ledge, a horizontal branch of a tree or suitable anthropogenic structures such as houses, water towers or bridges [1, 9, 16, 17]. A nest consists of a centrally located honeycomb covered on both sides with several layers most of the year by only female bee workers forming the so-called bee curtain [16, 18–23]. This formation is the main part of the mother nest of worker bees, and the roof-tile-like arrangement of the nest surface protects the nest interior with its resources from heat, cold, rain and wind [16, 24, 25]. The bees on the nest surface are also able to warn their nest mates on both sides of the comb of threats [26, 27], of troublemakers and especially nest predators [16, 24, 25] so that defensive measures can be initiated if necessary.

In an *Apis dorsata* nest, a mouth zone typically forms during the day [2, 28], which proves to be the interface between the inner and outer nest areas. This mouth zone has a specific, more intensive motion profile [28], because this is where forager bees leave the nest or return from visited nectar and pollen sources. There bees also exhibit dancing behavior [13] and trophallaxis [29–33], and when loaded with nectar or pollen, foragers find their way inside the nest here, and guards [34] also provide special protection from intruders here.

In the nest areas peripheral to the mouth zone, the nest members behave mostly *quiescent*, that is, they hang almost motionless fixing themselves with their extremities on the nest mates of the underlying layer, while they are oriented with the head upwards and the metasoma downwards [9, 14, 23, 35]. In particular, bees on this part of the nest surface may be disturbed in their quiescence by visual or mechanical influences and may be put into an alarm state. They can communicate this excitation to the other members of the nest by swinging their metasoma in upward direction [16, 23, 36–39]. Such a state of social arousal, preferably triggered by external visual stimuli, usually leads to a synchronised, collectively well-organised response pattern of many bees at the nest surface, which can drive away attackers, for example

predatory wasps, from the area of the bee nest [27, 40]. The excitation sequences on the nest surface between the individual bees, or also between different cohorts on the nest surface, are each associated with time delays in the millisecond range, which is visually perceived as a wavy shimmering pattern, at least for the outside observer [41].

The *semi-quiescent* state of the bee curtain, with motion activity in the mouth zone and quiescence in the peripheral nest areas, remains undisturbed most of the time, but is periodically interrupted during the day in steps of 3–5 hours by a colony-specific process that repeatedly reshapes the nest in its functional organization as a whole [42–44]. In this so-called periodic *mass flight activity* (MFA), almost all members of the bee curtain on both sides of the nest move around. They do this not only with their body parts, staying at their nest positions, but they also actively wander around on the nest surface, eventually flying away from the nest as well, forming a cloud of swarming bees around the nest for a few minutes [42–46] (S1–S4 Movies).

The magnitude of the MFA relates in good part to a periodic reorganization of the nest. It has the primary purpose of detaching the bees that were previously in the covering layer of the bee curtain, thus allowing them to feed and defecate. Within minutes this extensive motion activity begins to affect the entire bee curtain, which can be seen most clearly in the fact that the cover layer of the bee curtain is literally peeling off [42–44]. The other aspect of MFA concerns the preferably newly hatched workers, who can leave the nest like this for the first time, for their first orientation flights, but above all also to defecate. Nest-warm bees come from inside the nest through the net of curtain bees to the nest surface [14, 47] (see also S1 Movie), apparently independent of the normally established nest structure of the mouth zone and its periphery. After the end of the MFA, a new surface layer is formed again, whereby here participating bees perform their new social protective and defensive tasks, which are assigned to them according to their position on the nest.

Our hundreds of observations of MFA episodes on *A. dorsata* [9, 11, 35, 48] and *A. laboriosa* [49] colonies suggest that during MFA, the collective defense capability of the colony is undermined for at least 5–8 min. This tension between defensive readiness and MFA is a classic example of a *trade-off* situation [50–53], which is also proposed in this paper by the *mass-flight-suspend-defensiveness* hypothesis. This assumption is supported by the fact that during MFA the bees are fully engaged in rebuilding and restoring functional nest architecture and therefore hardly attend to external threats [42–44]. Such a state of reduced defensive capability is doubly dangerous for the colony, because on the one hand, nest predators would then have free and unhindered access to the resources in the nest, but what seems even more important, this risky state repeats itself several times a day, since such nest rearrangements must take place regularly and periodically. Potential nest predators would only need to wait for such low-risk moments as the MFA represents to plunder the resources of these nests. A bee colony that is in such a vulnerable state should therefore exhaust all possibilities so that precisely this long-term adaptation of nest predators to such opportunities favorable to the predators cannot take place.

In this paper, we investigate whether and how a colony in MFA mode, while reconfiguring its internal organisation, can maintain its defence capability, i.e. how the inhabitants on the nest surface are able to detect potential threats, alert the colony and initiate appropriate defence measures. The question here is also how long does such a dangerous emergency last for the colony, triggered by the out-of-question collective helplessness in the status of the MFA, and within what time can the colony regain its ability to defend itself?

For this purpose, the defensive readiness of an experimental colony was measured during several episodes of MFA. For this purpose, a dummy wasp was presented in cyclically recurring sessions directly in front of the experimental colony with a standardised movement

pattern. This made it possible to trigger shimmering behaviour and thus test the readiness of the colony to defend itself. Precisely because the dummy was presented regularly throughout the observation period, and was alternately before, during and after MFA, it is possible to also measure the change in the colony's defence readiness as a result of the MFA mode. For the evaluation, five topologically differentiated functional areas on the nest were defined and time profiles and spectra of motion activity were created from them. Arguments are being made as to whether the giant honeybees can perceive external threats collectively even in MFA mode, or whether they are even blind to external stimuli for a time. The criterion for this is the evidence that external threats could promote or inhibit motion activity in MFA mode.

## Materials and methods

### Experimental setup

**Study site.**  This paper relates to research on behaviors of defensiveness [16, 25, 40, 54] and mass flight activity (MFA) [42–44, 55, 56] in giant honeybees (*Apis dorsata*). From 28 Oct to 18 Nov 2010, we observed and videotaped around 80 episodes of MFA in seven different nests. For this case study, we selected three episodes (*mfa$_{1-3}$*) of a nest at *Eden Jungle Resort* in Sauraha (27˚34'28.4"N, 84˚30'01.4"E) with fully developed honeycombs, containing nectar, pollen and brood cells. For these three selected episodes only, we were able to partially or fully apply a defence readiness check programme (see below), and were also able to record the entire MFA episode, including a lead-in and a lead-out of 10 minutes each. The nest was attached to a concrete overlap of a balcony, allowing the equipment to be placed in front of the nest (Fig 1). The weather was sunny throughout the experiments, with daily temperatures from 19˚C in the morning and 30˚C at midday and a relative humidity of 67% - 78%. The meteorological data were recorded with two HOBO U12 data loggers at the observation site.

**Recording.**  The experimental giant honeybee nest was monitored with a Panasonic HVX-200 high-definition camera (Fig 1B). The '720p/50' mode used allowed recording with a spatial resolution of 1 280 × 720 pixels at 50 frames per s, for adequate documentation of motion activities on the nest such as flickering [57], shimmering [16, 26, 39, 40], dance behaviour [13] or locomotion [42]. Fig 1C shows the experimental nest just during MFA in episode *mfa$_1$*: Hundreds of bees buzz around the nest; their flight speed can be estimated from the length of their tracks during the recording time of 20 ms of a single frame. The geometrical arrangement of the bees on the surface of the nest here in the still image also indicates active motion activity, in MFA they are not as regularly arranged as in the state of semi-quiescence, where the bees hang like "tiles on a roof" in layers stacked on top of each other with exception of the mouth zone.

**Real-world calibration of the captured images.**  For the episodes *mfa$_{1-3}$* the image measures in the frames were set with 580 *px* width and 411 *px* height resulting in a conversion factor of 8.8 *px* /cm. Therefore, the real size of the experimental nest was determined in the field with the width of 66 cm and the height of 47 cm.

**Excitation of the experimental colony by the presentation of a dummy wasp.**  In this case study, a computer-controlled device with a dummy wasp was installed in front of the nest (Fig 1B and 1C; see the red arrows in Figs 2A$_{2a}$ and 3). This dummy served to sequentially trigger defensive behavior, such as shimmering waves [16, 27, 39, 57], in the colony under normative conditions. The wasp dummy consisted of a 5 cm long cylinder with yellow and black stripes, which was attached to the pulley with a thread in such a way that it could tumble back and forth during the horizontal motion (Figs 1C and 2A$_{2a}$). For episodes *mfa$_{1-3}$*, this dummy wasp was moved at a constant horizontal speed of 10 cm/s for at least 30 s at intervals of at least one min by passing parallel to the nest at a distance of 30 cm. These dummy movement

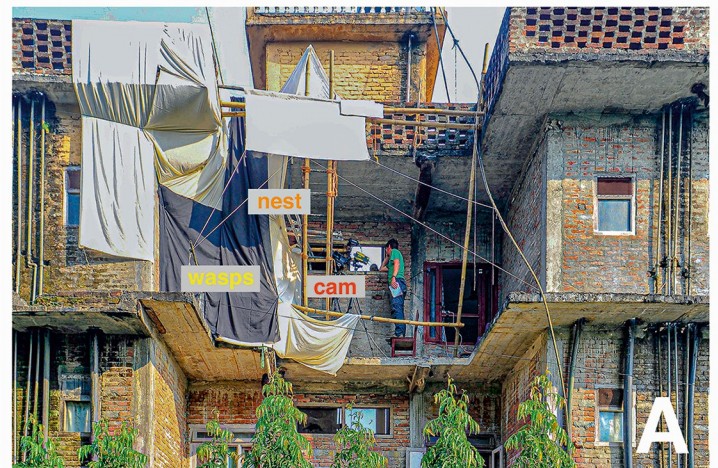

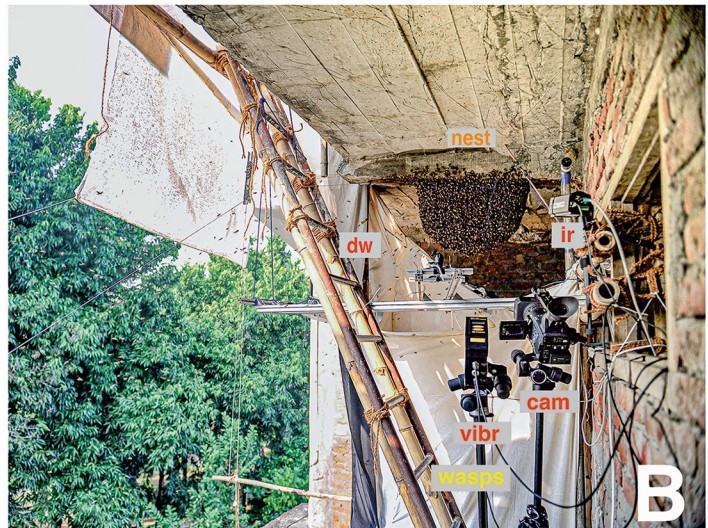

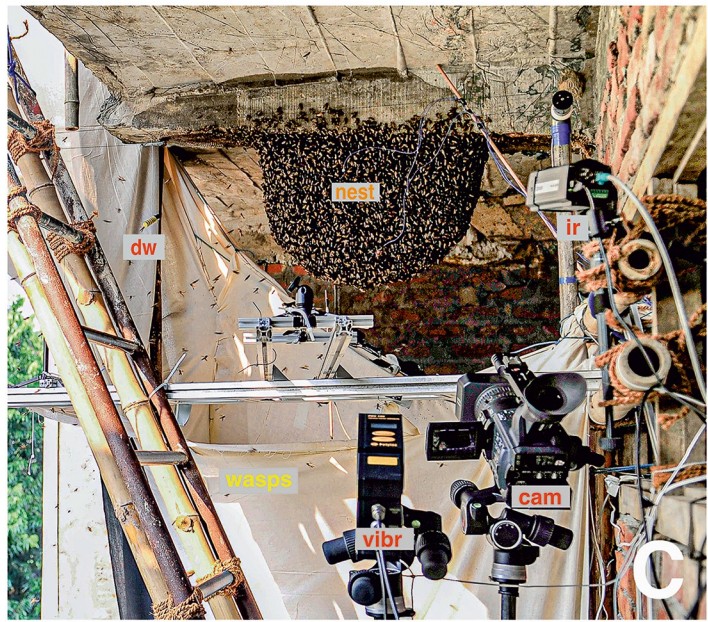

**Fig 1. Experimental site.** A, Experimental site on the unsecured balcony platform of the *Eden Jungle Res*ort in Sauraha (27˚ 34' 28.4 "N, 84˚ 30' 1.4 "E) with the measuring equipment and the experimental giant honey bee nest [*nest*], seen from the west. The wasp nest [*wasps*], which was brought to this location from several kilometers away, is protected from too bright sunlight with linen cloths. The technical equipment around the nest was arranged using struts and suspensions made of bamboo. B, the experimental setup seen from the south, with a video camera [*cam*], a computer-controlled device that moves a dummy wasp [*dw*] back and forth in front of the nest, an infrared camera [*ir*] and a Doppler vibrometer [*vibr*] (the latter are not dealt with in this paper). C, the experimental nest site in higher resolution during the mass flight episode $mfa_1$. The publication presented here refers to mass flight episodes alone, neither the wasp nest [27] nor the Doppler vibrometer [26] were addressed. The infrared camera (ir) was used to demonstrate collective ventilation [58], but also for a follow-up article [47] that is closely related to the one presented here on motion-activity data only. Reprinted from Photo Archive Gerald Kastberger under a CC-BY licence, with permission from Gerald Kastberger, original copyright 2010.

parameters were chosen to effectively trigger shimmering behaviour under nMFA (semi-quiescence; see below). This type of efficient stimulation was particularly evident in the fact that slightly higher velocities (> 20 cm/s) could even trigger mass recruitment [16] and the release of dozens of ready-to-sting flying guard bees.

**Experimental scheduling.** In this paper we investigate the question of how collective defence changes during the MFA of an *Apis dorsata* colony. To do this, it is necessary to start observing the nest in semi-quiescence before the MFA, preferably for at least half an hour, before the massive upheaval in the nest takes place and the individual bees are assigned new functions and positions. However, this also requires some experience with giant honey bees to be able to realistically estimate the start of such a process. In the field, we have been able to predict the onset of such an MFA with a span of one hour, taking into account comparable environmental and general weather conditions and especially the ambient temperature. It is also useful for a detailed analysis to observe the behaviour of the experimental colony for at least one hour after the end of the MFA. However, for practical reasons, the actual recording times were somewhat shorter than the actual observation time we spent at the nest.

## Definition of five surveillance zones on the experimental nest

For each of the three MFA episodes ($mfa_{1-3}$) selected for this paper, the collective motion activities were observed and measured in five surveillance zones ($sz_{1-5}$) on the nest surface (Fig 3), each defined in the image as a square area of 10 000 [*px*] with a side length of 100 *px*. Converted to the real world, each individual surveillance square corresponded to an area of 129.05 $cm^2$ with a side length of 11.36 cm.

The five surveillance zones represent five topographical areas of the nest where the bees involved are assigned different functions. Surveillance zone $sz_1$ [top left] was located just above the mouth area [2, 16, 35] of the nest and faced preferably the sun in the afternoon. Surveillance zone $sz_2$ [top-right] was located in the upper right area of the nest, it was farthest from the mouth zone and near the upper attachment of the nest to the concrete. At this point on the nest, the comb cells under the bee curtain were mainly used for storing honey and pollen rather than for rearing brood. Surveillance zone $sz_3$ [bottom-right] was located in the lower right nest region and thus also in the distant periphery to the mouth zone. It also touched, at least peripherally, the reproductive area centrally located in the nest. In this area ($sz_3$), the nest is continuously extended to the right and downwards by building new cells at the edge of the central honeycomb. Surveillance zone $sz_4$ [bottom left] concerned the mouth zone, which is the interface between the inside and the outside of the bee curtain. This is where most of the motion activities and social contacts that govern the collective life of the colony take place during the day; these are departure and arrival of the foragers, dancing [13], trophallaxis [29] and guarding [34]. Finally, the monitoring zone $sz_5$ [mid] was positioned in the central region of

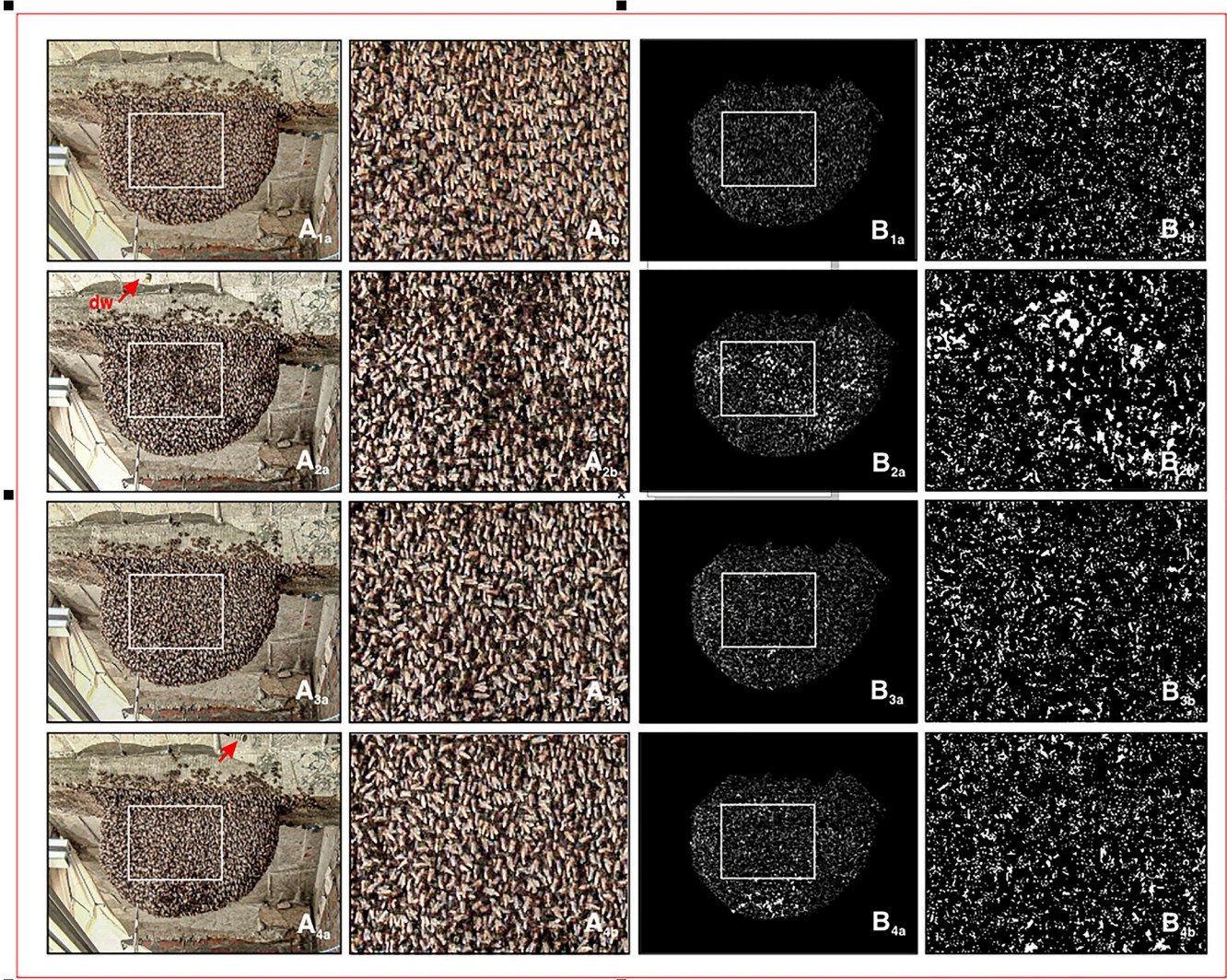

**Fig 2. Example images from the experimental nest during episode $mfa_1$.** Panels A, example original video images from episode $mfa_1$) regarding experimental sessions without presentation of the dummy wasp ($nP_d$), and with the dummy wasp presented ($Pd$) from nMFA and MFA states; panels B, the respective differential images (Eq#1a in S1 Text; see Methods). Here, motion activity detected at single pixel areas was represented as white pixel spots when the total RGB luminance of the respective pixel exceeded the threshold value $lum_{th}$ (Eq#2 in S1 Text). They appear brighter the more pixel areas are activated contiguously above threshold. Panels b contain in both image columns (A, B) those areas on the nest that are outlined in white in panels a. The small white arrow in panel $A_{1a}$ points to the white dot at the head of a stick poked through the honeycomb; this comes from the Doppler laser vibrometer (Fig 1C) which has measured the vibrations at the stick due to shimmering at the honeycomb [26]. The length of the stick protruding from the nest surface was used in a follow-up paper to measure the thickness of the layers of the bee curtain. Panels 1: $nP_d$ session under semi-quiescence (nMFA) without dummy presentation at frame f = 16 (at $t_{exp}$ = 320 ms after start of the observation); panels 2: $P_d$ session under semi-quiescence (nMFA) with dummy presentation (f = 1 216; 24.32 s); panels 3: $nP_d$ session under MFA (f = 12 507; 4.169 min); panels 4: $P_d$ session under MFA (f = 12 310; 4.1033 min). In front of the nest, a computer-controlled cable car-like mount with a dummy wasp ($dw$) was placed to provoke the defensive behavior of the colony. This was a black and yellow striped cylinder suspended by a thread 30 cm in front of the nest, which could be pulled back and forth (marked by the red arrows in $A_{2,4}$). In this perspective, the camera captured the dummy from a more bottom angle so that the dummy appears here at the top of the picture. Reprinted from Photo Archive Gerald Kastberger under a CC-BY licence, with permission from Gerald Kastberger, original copyright 2010.

the nest. In this nest area, directly under the bee curtain on both sides of the central comb, is the main reproductive area of the colony. This surveillance zone borders on the left on the zones of the left side of the nest ($sz_{1,4}$) and on its right on the edge zones $sz_{2,3}$ and is located in the vertical center between the upper attachment zone and the lower nest rim.

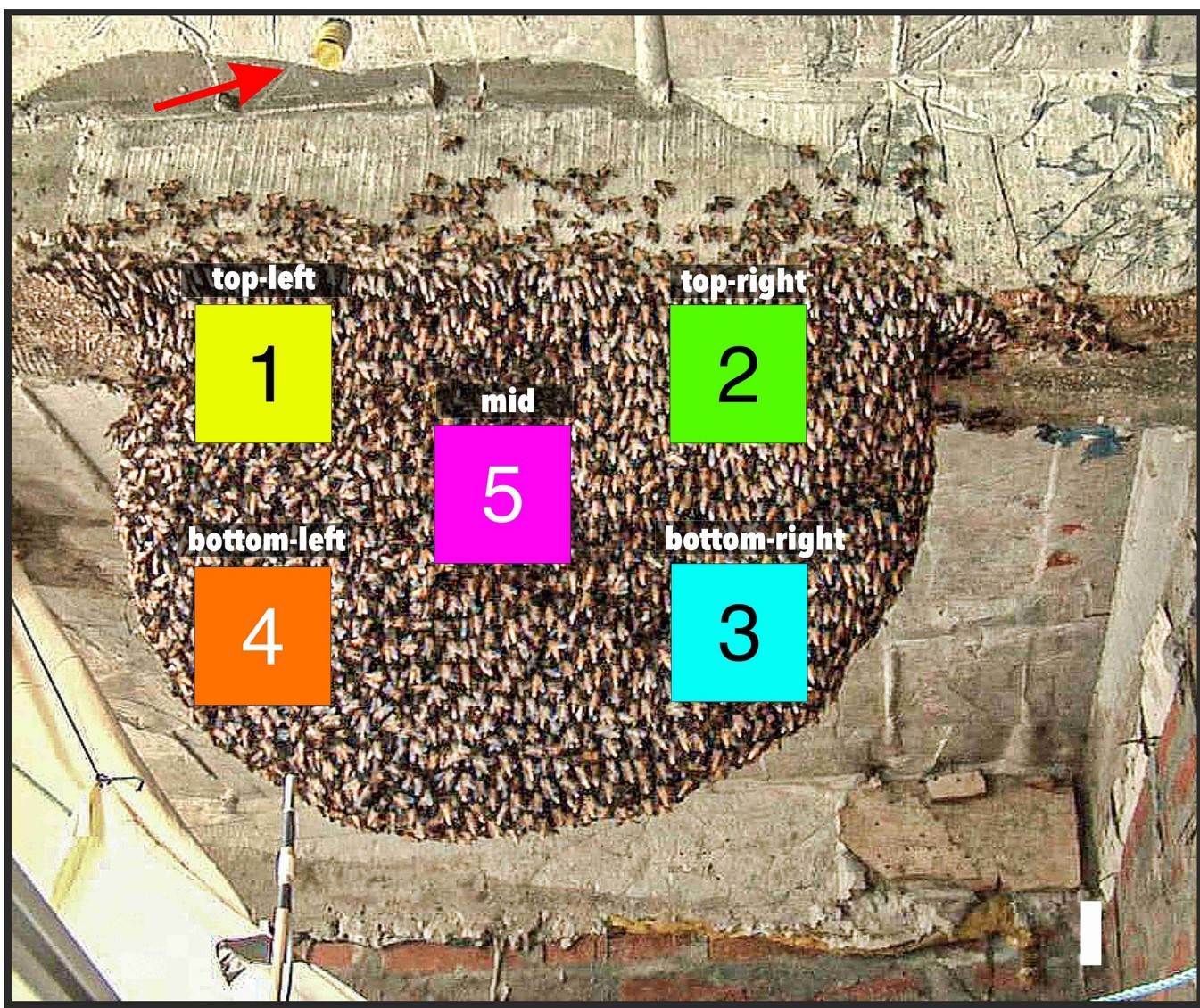

**Fig 3. Five surveillance zones at the experimental nest.** Surveillance zones were set up to track motion and heat activities [47] at the experimental nest during episodes $mfa_{1-3}$ at five functionally different locations. They were each defined in the image as a square area of 10 000 [px] with a side length of 100 px. Converted to the real world, each individual surveillance square corresponded to an area of 129.05 cm$^2$ with a side length of 11.36 cm. Surveillance zone $sz_1$ (top left) is located on the left side of the nest, which was also regularly illuminated by the sun, and was located above the mouth zone; $sz_2$ (top right) is to the side near the wall of the hotel and near the attachment zone to the ceiling; $sz_3$ (bottom right) is located below $sz_2$ and borders the lower edge of the nest; $sz_4$ (bottom left) represents the mouth area where there was busy traffic of departing or arriving foragers during the day, guard bees patrolled here and controlled the arriving bees, trophallaxis took place with arriving bees providing nectar for the recipient bees, and finally dancers were active here to provide and disseminate information about the effective food sources. Finally, $sz_5$ (nest center) is located in the center of the nest, where the reproductive cells are located underneath the bee curtain on both sides of the honeycomb; $sz_5$ also touches the edge of the mouth zone, so it is among those regions where most of the shimmering waves were initiated [28]. The colour codes listed here are used to identify the respective surveillance zones in all other figures. The red arrow marks the dummy wasp with yellow and black stripes, which was attached to a thread and moved back and forth in front of the nest by remote control. As the camera was pointing slightly upwards towards the nest, the dummy is shown here at the top of the picture.

## Assessment of motion activity in surface bees

In a giant honeybee nest, a range of individual and collective behaviors are associated with *motion*. In this paper, the term *motion* is used to describe spatio-temporal changes of bees at the nest surface that can be observed and measured over time from frame to frame. *Motion*

refers to the dynamic postural changes of these surface bees, including small twitches of body parts, parallel shifts of the entire body conditioned by the slight changes in the positions of neighbors, and most importantly, it also refers to the flinging up of the metasoma in the active state of shimmering [23, 26, 38, 39] or fanning at the spot [58]. This paper explicitly distinguishes the *movements*, which are equated here with locomotion. This involves crawling on the surface of the nest, flight maneuvers such as taking off or landing, or the agents emerging from the subsurface layers of the bee curtain, or vice versa, disappearing from the surface into deeper layers (S1–S4 Movies).

The individual shimmering pulses of each agent last up to 300 ms, with the metasoma being flung up within 70 ms and passively dropped in the further 200–300 ms [23, 39]. Overall, the collective of surface bees can wave the entire nest like a curtain through their synchronized but also cascading triggering of shimmering actions and even stimulate the central honeycomb to vibrate [26] especially because such shimmering actions are generated in cycles of about 1 Hz. Consequently, if the nest is recorded with a fixed camera, *motions* and *movements* lead to fluctuations of the luminance values at the corresponding pixel [*px*] positions (Eq#1, #2 in S1 Text).

**The differential luminance as a parameter for motion.** Motions on the surface of giant honeybee nests were quantified using image analysis (Image-Pro®, Flir; Fiji [59]) by determining differences in luminance values pixel by pixel between two consecutive images (Eq#1 in S1 Text). However, such fluctuations in pixel-related luminance values can also occur due to brightness fluctuations in the environment (e.g. when clouds move in front of the sun, which can also happen quite quickly). Such changes in luminance values occur in the order of minutes and also usually affect the entire nest area uniformly. In contrast, *motions* during shimmering or *movements* during honeybee locomotion on the nest surface, are associated with much faster luminance changes that occur mostly in fractions of a tenth of a second and are locally specified. Since changes in ambient brightness do not seriously affect luminance-based monitoring of motion at the nest surface, it can be assumed that any significant change in luminance that can be detected in a single pixel area between two successive images in the documented sequences (Eq#1 in S1 Text) was caused by individual or collective, active or passive motion.

**Detecting motions.** The high resolution of differential luminance at the pixel level, defined in 24-bit RGB format, enables the detection of extremely small magnitude motions, even if they only occur on certain parts of a single agent's body. The entity of all image-based pixels of two successive frames forms a differential image at the time point $t_i$ (Eq#1; Fig 2B). The difference images were digitally filtered, that is, they were segmented with respect to the luminance threshold $lum_{th}$ = 10 (Eq#2 in S1 Text). According to this concept, a single pixel area [*px*] that has a differential luminance value above the threshold is assigned the state *motion-active*, and pixels with subthreshold differential luminances are assigned the state *motionless*. To still minimize ambient noise in the difference images (Fig 2B), they were further refined using the digital pixel filtering functions of *erosion* and *dilation* [59].

As a rule, *motion-active* pixels were not randomly distributed in the difference images, so they always manifested as distinguished zones of motion activity. Therefore, the motion activity of the colony in a given region on the surface of the nest is also well quantifiable with the measured quantity (Eq#3a in S1 Text) at the given time $t_i$ for the given spatial pixel resolution. For this purpose, the *motion-active* pixels in a defined reference region of the nest were counted. This can be the entire nest itself or only subregions of it, as also shown for the five surveillance zones $sz_{1-5}$ (Fig 3). For each MFA episode ($mfa_{1-3}$), the maximum motion activity was determined, taking into account all surveillance zones over the entire time range from the first to the last frame of the analysis section of the recording (Eq#3b-c in S1 Text). Finally, the

normalized quantity of motion activity (Eq#3d in S1 Text) allows the comparability of the motion strengths per session because of the different durations.

### Establishing four states of excitement in the experimental colony

**Semi-quiescence (sQu respectively nMFA).** For most of the day, a giant honeybee colony shows a restful, *quiescent* state, with only a small number of bees in motion, which are found in particular in the mouth zone [2, 16], represented by the surveillance zone $sz_4$ (which can be detected with naked eyes in Fig 2A$_1$ and 2B$_1$), but much less in the regions peripheral to the mouth zone, represented by the surveillance zones $sz_{1-3,5}$ (Fig 3). The term *semi-* is used here because most of the nest is in *quiescence*, while motion, at least in parts of the agents' bodies. or even locomotion is still continuing in the mouth zone.

**Arousal triggered by a dummy wasp.** The experimental colony was repeatedly excited in a specific cycle for minutes by the recurrent presentation of a dummy wasp (Figs 1B, 1C, 2A$_{2,4}$ and 2B$_{2,4}$). Sessions without the presentation of the dummy wasp (nP$_d$) regularly alternated with those in which the dummy wasp was presented (P$_d$) for at least 30 seconds. For instance, in the episode *mfa$_1$* (Figs 4A$_{1-5}$ and 5A$_{1-5}$) the P$_d$ sessions lasted around 1 min ($\Delta t$ [P$_d$] = 1.018 ± 0.059 min; n = 8), while the nP$_d$ sessions had a duration of about 2 min ($\Delta t$ [nP$_d$] = 2.166 ± 0.209 min; n = 8).

Therefore, in the example of the episode mfa$_1$ the status of quiescence (nMFA) can be assigned for surveillance zones with the exception of the mouth zone ($sz_{1-3,5}$) for the session nP$_d$ 1 before MFA started, and also for the sessions after the termination of MFA, nP$_d$ 9–17 with odd counts (Fig 4A). The status of the mouth zone ($sz_4$) is non-quiescent due to the loco-motor activities as long the mouth zone exists during the day (sometimes the giant honeybees collect nectar and pollen even at night time [60, 61]). Non-quiescence in the surveillance zones, which are peripheral to the mouth zone, is formally characterized by two criteria: firstly, by the occurrence of shimmering behavior in response to the presentation of the dummy wasp, which happens in the P$_d$ sessions predominantly in nMFA state; and secondly, the col-ony-specific disorder of nestmates during MFA, which affects all zones and layers of the bee curtain, even on both sides of the nest.

Since the observation windows for episodes *mfa$_{1-3}$* cover not only the MFA period, but also the nMFA periods before and after, four categories of excitation regimes can be distinguished (as shown in panels 6 of Fig 4): (a) the state of *semi-quiescence* without excitation by the dummy presentation [nP$_d$, nMFA]; (b) the nMFA state with the excitation by the dummy pre-sentation [P$_d$, nMFA]; (c) the state of mass flight (MFA) without excitation by the dummy pre-sentation [nP$_d$, MFA]; (d) the state of mass flight with dummy presentation [P$_d$, MFA].

In contrast to *motion profiles*, *motion spectra* are distributions of motion data that were col-lected in a certain time range in the observation window and are sorted according to motion strengths. In this work, *histogram-based* spectra are used, whereas the raw motion data (Eq#3a in S1 Text) are sorted along a size scale. Therefore, the number of cases per size class is plotted on the ordinate, and the latter is considered on the abscissa. Ordinate and abscissa values of the spectra are normalized in order to be able to compare between different experimental conditions.

**Detection of low-frequency components in the motion sequence in a semi-quiescence session.** For the detection of a low-frequency event immediately before the actual start of the MFA (e.g. nP$_d$ 3; *mfa$_1$*), the motion sequences were analyzed with an algorithm substituting a Fourier transform for the wave preference analysis (for the complete algorithm, see Eq#4 in S1 Text). The database of these charts considered the five surveillance zones ($sz_{1-5}$) only in the epi-sode *mfa$_1$* and because the nP$_d$ 3 session lasted 7 994 frames, this analysis included 39 970

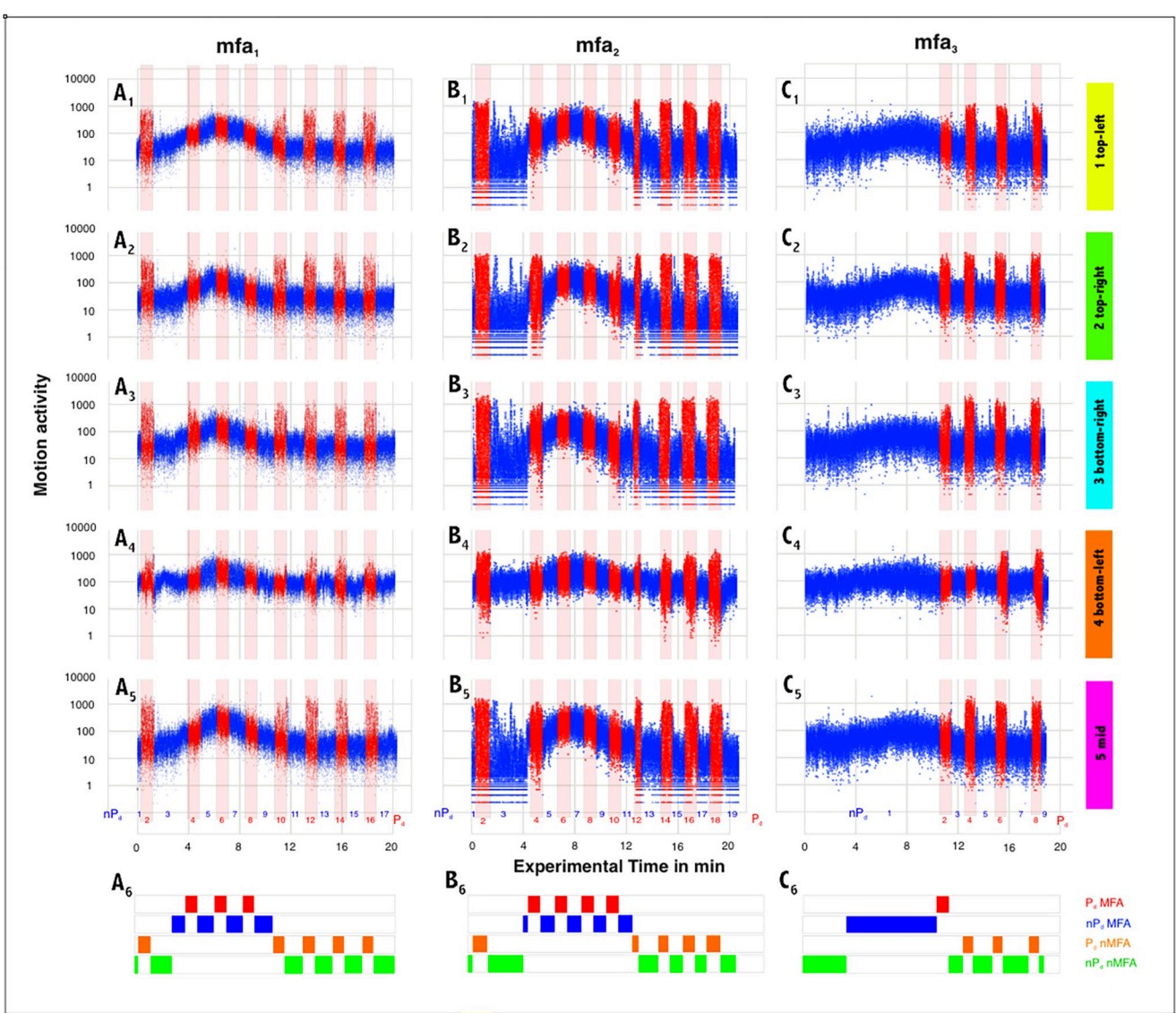

**Fig 4. Motion profiles from three episodes (*mfa$_{1-3}$*) in five surveillance zones (*sz$_{1-5}$*) defined on the experimental nest.** Panels A-C show episodes *mfa$_{1-3}$* each ranges over 17–19 sessions with alternating arousal levels (red: P$_d$ with dummy wasp presented; blue: nP$_d$ without the presentation of the dummy); panels 1–5 refer to the surveillance zones *sz$_{1-5}$* as tagged with the color bars at the right side (corresponding to the schematics in Fig 3). Ordinates (A-C): the number of motion-active pixel areas [*px*] within a single frame (Eq#3a in S1 Text); since the surveillance zones (*sz$_{1-5}$*) in the frames have a size of 100 x 100 px, the theoretically maximum possible motion intensity per frame would then be reached if the full number of 10 000 [*px*] each were designated as motion active (Eq#2 in S1 Text). Abscissa: the experimental time in minutes. Each plot comprises > 60 000 data points. Panels 6 give the survey about the four categories of experimental sessions: green, nP$_d$ nMFA; orange, P$_d$ nMFA; blue, nP$_d$ MFA; red, P$_d$ MFA.

records for all five surveillance zones. Differences in motion activity for a given surveillance zone (*sz$_{1-5}$*) were determined for eight time intervals between successive images, which then ranged from 0.5 to 50 Hz when converted to the frequency domain. Finally, the normalized motion values were entered in the frequency diagram (Fig 6 and 6B) and can be used as a guideline to decide whether, to what extent and in which nest area (*sz$_{1-5}$*) any crucial low-frequency components at frequencies of 0.5 Hz or even lower have occurred in the session nP$_d$ 3 at the beginning of the MFA (*mfa$_1$*).

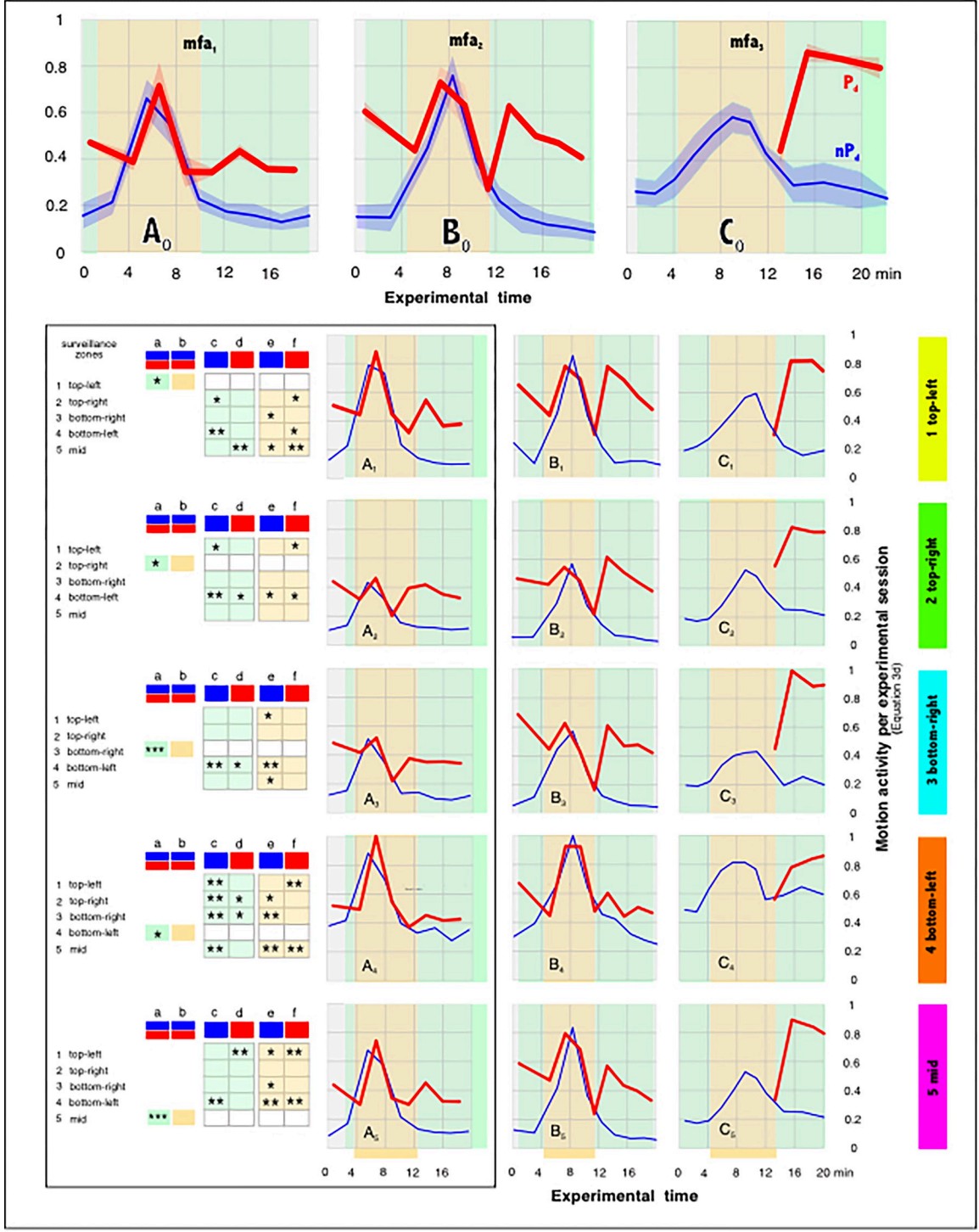

**Fig 5. Motion profiles of the episodes *mfa₁₋₃* for each of the five surveillance zones *sz₁₋₅*.** Ordinates, benchmarks of motion activity (Eq#3d in S1 Text) for the experimental sessions in a selected surveillance zone ($sz_{1-5}$: panels 1–5 as tagged with the color bars at the right side corresponding to the color coding in Fig 3) of a selected mass flight episode (A-C: $mfa_{1-3}$); abscissas, the experimental time in min; $P_d$ sessions (red lines) and $nP_d$ sessions (blue lines) as displayed in Fig 4A–4C. Episode $mfa_1$ (A): 9 $nP_d$ sessions | 8 $P_d$ sessions |$n_{ff}$ = 60 713 frames; $mfa_2$ (B): 10 | 9 | 63 340; $mfa_3$ (C): 12 |4 | 68 320. Bright green background areas, state of semi-quiescence (nMFA) before and after MFA; bright orange background areas, state of MFA. $A_0$-$C_0$ show the means (solid lines) and mean errors (transparent clouds above and below the means) of the benchmarks (Eq#3d in S1 Text) considering all five surveillance zones ($sz_{1-5}$). The episodes $mfa_{1-2}$ display sequences alternating between $nP_d$ and $P_d$ states; in the episode $mfa_3$ the dummy wasp was presented only after the end of MFA.

For that, the experimental $nP_d$ sessions for the time previous to the $P_d$ phases was portioned in steps of 5 000 frames. Panels at the left side summarize the results from t-tests (*, P < 0.10; **, P < 0.01; ***, P < 0.001) to compare the motion activity spectra only for the episode $mfa_1$ (Panels A) between $nP_d$ (blue) and $P_d$ (red) conditions for each of the surveillance zone ($sz_{1-5}$): within each surveillance zone (tests a,b), and between the surveillance zones in the $nP_d$ sessions (test c: nMFA; test e: MFA) and the $P_d$ sessions (test d: nMFA; test f: MFA).

### Histogram-based motion spectra

In principle, motion activity of the surface bees is allowed to be viewed with a spectral measure, according to classical mechanics of nonlinear oscillators [62, 63], ranging from micro changes in their position (e.g. when they are pushed by neighbouring bees) to massive changes in their posture and position when they show shimmering behaviour, or when they move from their position. In particular, if this motion parameter is defined by the changes in luminance (Eq#1–3 in S1 Text) in the associated areas of the images acquired over a period of time, the sequentially measured motion activity can be considered as a continuous spectrum. In the histogram-based plot, for each experimental session ($P_d$, $nP_d$) of an experimental episode ($mfa_{1-3}$), the image-based motion activity of the respective surveillance zone ($sz_{1-5}$) was quantified by the number of motion-active pixels per image (Eq#3a in S1 Text), and sorted into one of 50 logarithmically scaled size ranges (Eq#5a in S1 Text). To keep the sessions with the different arousal states comparable across the resulting number of cases per interval of motion strength, the values of motion activity per session were normalized twice: first by the number of frames of the respective session (Eq#5e in S1 Text), as session durations differ, and second by the maximum value of the entire episode (Eq#5g in S1 Text) to achieve a relative scaling between the normalized minimum (equal to value 0) and the maximum (equal to value 1) for each episode.

**Evaluation of the skewness of the histogram-based motion spectra.** The logarithmic scaling of the motion strength (Eq#5a in S1 Text) gives the motion spectra in semi-quiescent state an approximately Gaussian distribution pattern (Fig 7). This effect is understandable because often the linear neural control of muscles responsible for movements typically leads to an exponential change in effective force (see textbooks on muscle physiology). The quotient $q_s$ (Eq#6g in S1 Text) is here introduced as the measure of skewness which gives the ratio of the areas of the right and left tails of the histogram-based motion spectra (Eq#6e-f in S1 Text). This parameter can be considered very useful in deciding how the nestmates of a colony collectively change or even lose their sensitivity to external threat signals when they enter MFA.

For example, if this $q_s$ value remains almost unchanged from one to the other $P_d$ session, it can be assumed that the bee cohorts in the corresponding nest area have also remained similarly sensitive with regard to the perception of the visual threat signal. On the other hand, turning the motion profile asymmetric ($q_s \neq 1$) indicates change in the sensitivity to external stimuli, which with the condition $q_s > 1$ signals a dominance at the right end of the motion spectrum; this is true when cohorts in $P_d$ sessions were stimulated to perform shimmering actions under semi-quiescent conditions (e.g. Fig 7B: $P_d$ 2; $sz_1$ [top-left]). In contrast, the condition $q_s < 1$ leads to a left-skewed spectrum (e.g. Fig 7B: $P_d$ 6; $sz_4$ [bottom-left]). Asymmetric spectra ($q_s \neq 1$) must first be seen in the context of being caused by both the MFA state itself and the dummy presentation. Mostly, it can be assumed that such skewness hides an attenuated bimodality caused by two separate cohorts.

**Base motion activity during mass flight in comparison between episodes.** The histogram-based motion spectra (Eq#5 in S1 Text) also allow the calculation of the base motion activity of an experimental session. This is done by normalising the respective peak of the MFA over several thousand frames. From both episodes under investigation (e.g., $mfa_1$ and $mfa_3$), the histogram-based motion spectra are normalized to the respective peak of MFA over

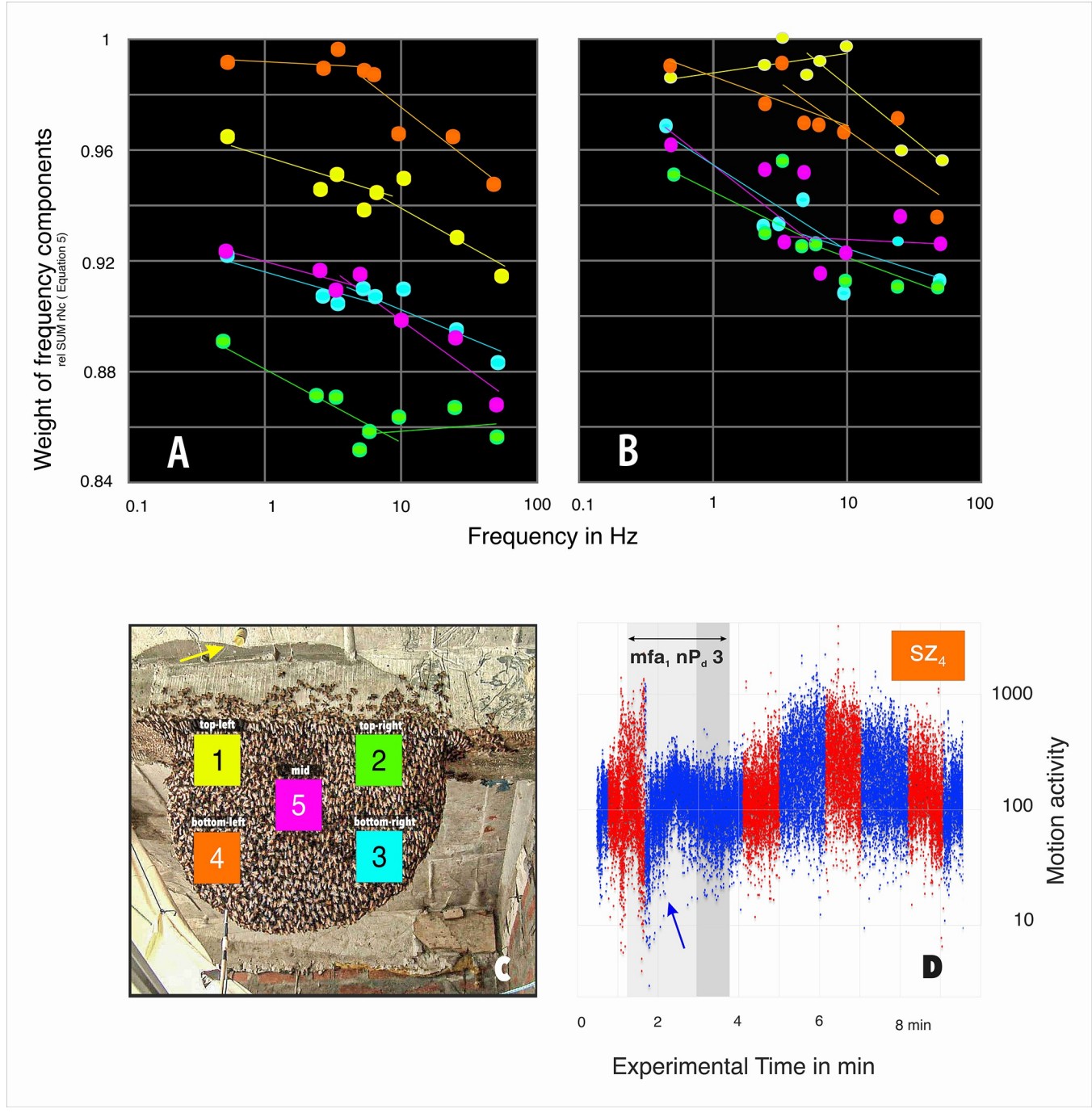

**Fig 6. Do low-frequency ripples in motion profiles (nP$_d$ 3 phase; *mfa$_1$*; *sz$_{1-5}$*) signal the onset of MFA?.** A,B: These plots refer to the nP$_d$ 3 session of the episode *mfa$_1$* in which the nest shows marked unrest as a low-frequency signal in motion sequence. In Panel D, this part of the motion sequence is displayed more enhanced for the example of *sz$_4$* (for the complete motion sequences, see Fig 4A–4C), the experimental session nP$_d$ 3 is marked by grey underlays, whereby both phases of this experimental session have been considered separately (D: phase a, bright grey, represented in panel A; phase b, darker grey, represented in panel B). The points in panels A,B are marked by the color of the respective surveillance zones *sz$_{1-5}$* (as defined in panel C); ordinate: the parameter *weight of frequency components* (according to Eq#5e in S1 Text); abscissa: the reciprocal of the time interval between selected frames scaled in Hz. The intervals selected for this on the motion activity timeline range from 1 frame to 100 frames apart.

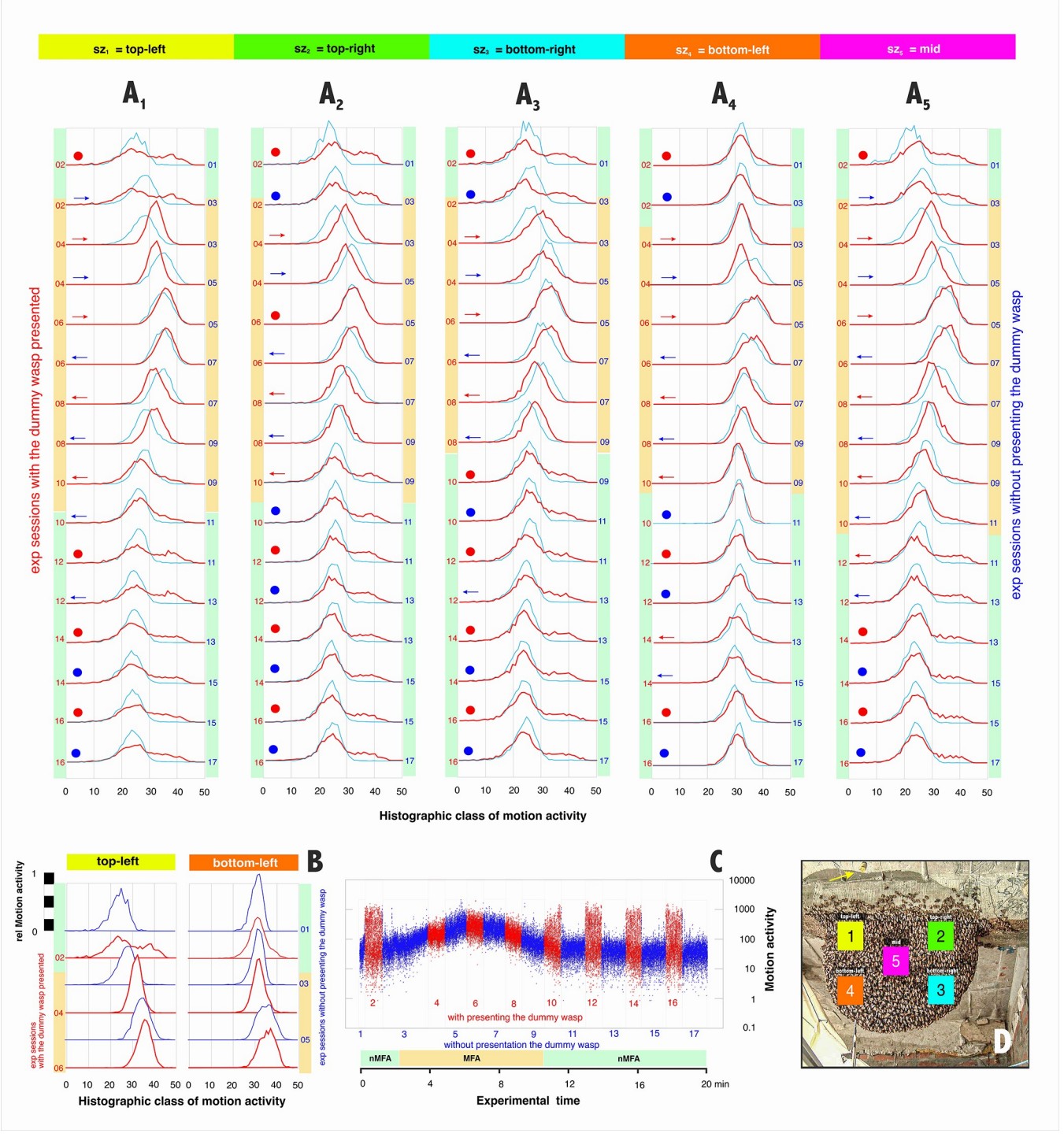

**Fig 7. Histogram-based motion spectra of episode *mfa₁*.** A, the motion spectra of two surveillance zones ($sz_{1,4}$; for color identification, see Panel C) for each of the 17 experimental sessions of $mfa_1$ (see panel B for session identification). Ordinate: The number of images in which motion activity (Eq#2 in S1 Text) was detected in each surveillance zone in one of 50 categories plotted as intervals of motion intensity on the abscissa (Eq#5 in S1 Text). In each of the line graphs in Panels $A_{1-5}$, the motion spectra of two consecutive sessions are shown. This facilitates the assessment of changes in the spectra from session to session, which are arranged vertically along the time axis. The individual spectra can be identified by the session numbers (red, $P_d$ sessions, each on the left-hand side; blue, $nP_d$ sessions, each on the right-hand side), and related to the corresponding behavior category (see bars on the side of the diagrams: green, nMFA semi-quiescence; orange, MFA). Horizontal blue and red arrows at the left side of each column indicate the direction in which the corresponding motion spectrum has shifted along its abscissa, i.e. in the ascending MFA the spectra drift to the right, in the descending MFA shift to the left. The red and blue full circles indicate that the respective spectra remained unshifted. B, sample spectra for further explanation, with the black & white scale displayed at $nP_d$ 1 ($sz_1$). This

value was normalized between the values 0 and 1 by the number of frames of the respective experimental session and by the maximum value of episode $mfa_1$ (Eq#5g in S1 Text) in order to compare all spectra of episode $mfa_1$. C, dot plot of a motion profile ($mfa_1$ / $sz_1$, same as Fig 4A$_1$) associated to the spectra of panel A$_1$. Ordinate: motion intensity (Eq#3a in S1 Text) of the sequential frames (red, P$_d$; blue, nP$_d$) against the experimental time in min (abscissa). The log motion scale illustrates that the mass flight phase lasts over 10 min, documented by the restlessness of the collective motion which appears as a slight peak in the red coded P$_d$ and the blue coded nP$_d$ sessions. D, sample image of episode $mfa_1$ in semi-quiescence phase P$_d$ 2 with the definitions of the surveillance zones $sz_{1-5}$. Here the dummy wasp was presented (see yellow arrow on top of the image), which triggered a shimmering wave that spread over the nest.

several thousand frames (according to the Eq#5 in S1 Text). Lastly the maximum value (Eq#5f in S1 Text) was assessed regarding all experimental sessions (e.g. sess$_{exp}$ = 1..17 for $mfa_1$), and for all surveillance zones ($sz_{1-5}$), but separately for the three episodes mfa$_{1-3}$ (Fig 4A–4C). This episode-specific peak value was used to normalize the basic motion data to range between 0 and 1 (Eq#5g in S1 Text) for easier comparison.

Subsequently, the class measures are de-logarithmised (Eq#7b in S1 Text) to form the products of the motion magnitude pairs and the relative class-specific number of cases, and thus all partial products are summed up (Eq#7c in S1 Text). If one forms the quotient (Eq#7d in S1 Text) between the two cases (Eq#7c in S1 Text) over e.g. $mfa_1$ and $mfa_3$ and takes the rel fraction of the episode $mfa_1$ in the denominator, the rel fraction Q $_{mf}$ (Eq#7d in S1 Text) with respect to the basic motion activity under nP$_d$ conditions in MFA is obtained for the episode $mfa_3$. However, one problem arises when comparing two episodes because the results of image analysis cannot be directly compared due to probable episode-specific resolution sensitivity for motions. As in the present work demonstrated, this can be compensated for by considering the motion activities of both episodes only in relation to their respective maximum P$_d$ and nP$_d$ sessions.

## Matching the empirical skewness data with two motion inhibition models

In the *progressive-inhibition-during-mass-flight* (*pim*) hypothesis, the motion data of skewness of the histogram-based spectra (parameter $q_s$; Eq#6g in S1 Text) decreases linearly from the maximum at the beginning of the MFA to the minimum at the end of the MFA (Eq#8a-c in S1 Text). This means that the portion of shimmering ability is correspondingly reduced in the course of MFA. At the end of MFA, this measure switches back in its value to the maximum. In this work, the alternative *inhibition-proportional-to-mass-flight* (*ipm*) involves a "semi-circular" depression from the beginning to the end of the MFA (Eq#8d-h in S1 Text), representing a non-linear, gradual decrease in shimmering activity at the beginning of the MFA and a gradual re-increase at the end of the MFA. After normalizing the empirical skewness data between the values 0 and 1 (Eq#8i in S1 Text), the data are fit for comparing them with both models. The respective match is calculated by the Pearson correlation coefficient (PCC: Eq#7j in S1 Text).

## Definition of trade-off parameters with measures of motion activity

In giant honeybees, the readiness to defend is obviously antagonistic to the state of MFA, it is simply difficult to manage both tasks simultaneously. This aspect resembles a trade-off [49, 50, 63] relationship, where two variables are in an antagonistic relationship that can be mathematised as a hyperbolic (Eq#9a in S1 Text) function, normalised between 0 and 1, and linearly transformed by double logarithmic scaling into a diagonal extending from [0,1] to [1,0] at an angle of -45˚ (Eq#8b in S1 Text); however, this only applies if the coefficient A has the value 1 (Eq#9a in S1 Text).

In this paper, a suitable pair of parameters was extracted from the histogram-based (Fig 7; Eq#10 in S1 Text; see also definition in Fig 10D) motion spectra to quantify such an

antagonistic relationship. The first parameter describes the *state of the MFA* (abscissa values in the plots of Fig 10) by the intensity of collective motion activity in a surveillance zone ($sz_{1-5}$), with the ability to distinguish the motion state of the MFA from that under semi-quiescence. The second parameter quantifies the *state of defensiveness* (ordinate values in the plots of Fig 10) by indicating the extent to which shimmering responses are triggered.

The instantaneous state of MFA was quantified by the relative position of the peak of the Gaussian distribution of the respective $nP_d$ session on the abscissa of the motion spectra (Figs 7A–10E; Eq#10b in S1 Text). This value is lower in nMFA and increases during MFA. On the other hand, the instantaneous defensiveness, i.e. the extent to which shimmering responses can be triggered, is quantified by the $q_s$ value (Eq#6g,#10a in S1 Text) of the motion spectrum of any $P_d$ session following the preceding $nP_d$ session, which in turn serves as a measure of MFA status.

In a second, additional method, the two antagonistic measures were extracted from quantile-based spectra (see Eq#11–13 in S1 Text). Here, the defensiveness was quantified by the mean slope of the quantile-based spectrum in the selected part from 0.30 to 0.90 (Eq#12 in S1 Text) and the MFA state was quantified by the low base of the quantile-based spectra (Eq #13 in S1 Text).

## Statistics

Basic statistical operations are used to describe time courses, t-tests to distinguish Gaussian distributed data between different behavioral conditions, and Pearson correlation coefficients as tests to prove significance in correlations.

## Ethics statement

The research expedition to Chitwan, Nepal, entitled "Study on the behavior of the giant honeybees: Observations and recording of behaviors at the nesting site" was supported by the Rector of the Centre for International Relations of the Tribhuvan University (Kathmandu, Nepal).

## Results

### Monitoring mass flight episodes

An *Apis dorsata* nest can go from a state of semi-quiescence (sQu) to mass flight (MFA) within a few minutes [42–46]. This is shown by the time profiles of the motion activity (Eq#3a in S1 Text) of the bees at the nest surface, plotted separately for the five observation zones $sz_{1-5}$ (Fig 4). The transition from sQu to MFA can be seen in the plots of the blue dots of the three episodes $mfa_{1-3}$, which shift to larger amplitudes and form a peak for 6–8 minutes.

In these experiments, a dummy wasp was presented in front of the nest at regular intervals in the Pd sessions, in contrast to the $nP_d$ sessions, in which no dummy was presented (the entire episode $mfa_1$ is available in the Supplementary Material as a representative of all episodes $mfa_{1-3}$ analysed in this paper: S1 Movie–S4 Movie with the comments in S2 Text; the sequence of motion activity is visualised here for the surveillance zone $sz_1$). This stimulation triggered arousal with additional locomotor activity, leading in particular to shimmering behavior [16, 26, 28, 38–40], which by default spread in waves throughout the nest.

In the episodes $mfa_{1-2}$, the $P_d$ and $nP_d$ sessions yielded similar motion patterns. Before and after the MFA, the motion activities in the $P_d$ sessions (Fig 4A and 4B) were higher than those in the $nP_d$ sessions (blue coded), which is indicative of the shimmering activity. Once, at the beginning of the MFA, the $nP_d$ data have increased to the motion level of the $P_d$ data (Fig 4A and 4B), both conditions showed similar motion patterns, which means that in MFA mode the

surface bees' defense reaction to the wasp dummy is limited or even completely prevented. After the peak of the MFA episodes, the motion data decrease again in both session types ($nP_d$, $P_d$) until finally that motion level is reached at which the readiness to shimmering is switched on again in the $P_d$ sessions.

This general description applies in principle to all five observation zones $sz_{1-5}$, although there are some important differences. One of these concerns the level of motion activity in the $nP_d$ sessions, which is significantly higher in the nest areas at the mouth zone ($sz_4$) than in the more distant nest zones ($sz_{2,3}$). This is due to the ongoing locomotion activity of the bees, which are located in the mouth zone.

Episode $mfa_3$ (Figs 4C and 5C) brings another aspect here, showing how the motion pattern develops during MFA without being disturbed by a cyclic presentation of a dummy wasp. In this case, the motion curves in MFA remained significantly lower than in the two episodes $mfa_{1,2}$ (Fig 4A and 4B). Although the motion amplitudes were normalized separately in each of the episodes (Fig 5), the relationship between the peak of the $nP_d$ curves in MFA and the height of the $P_d$ curves in subsequent sQu concerning the experimental time of >12 min shows that the latter are much larger than in the episodes $mfa_{1,2}$. Mean motion activity from all five surveillance zones ($sz_{1-5}$) during the peak of MFA is half as high when calculated with the factor $Q_{mf}$ = 0.47179 ± 0.03619 (x ± sx; $n_{sz}$ = 5; Eq#7d in S1 Text) for episode $mfa_3$, where the cyclic dummy presentation occurred only after the end of MFA, in contrast to episodes $mfa_{1,2}$.

## A signal for starting mass flight?

In the immediate run-up to MFA, before the baseline activity of the collective motion had increased under sQu (Fig 4A–4C), a characteristic ripple appeared over 1–2 min in the motion sequence. In both episodes $mfa_{1,2}$, this can be observed especially in the first section of the $nP_d$ 3 sessions (Fig 4A$_4$ and 4B$_4$), but such ripples are also present in the otherwise less disturbed course of episode $mfa_3$ (Fig 4C$_4$). In Fig 6A and 6B this motion turbulence was identified by its frequency components (Eq#4e in S1 Text) in the range from 100 Hz down to 0.1 Hz, and was found to be distributed in a specific topographic pattern over the nest surface. The highest level of motion of these frequency spectra was detected in the mouth zone ($sz_4$), confirming the respective manual observation in Fig 4A$_4$ and 4B$_4$. According to the maximum components of these waves, the order of the monitored nest areas was as follows: at the top of this order was the area above the mouth zone ($sz_1$), just below that the nest centre ($sz_5$), and finally followed by the peripheral nest areas ($sz_{2,3}$); here, the upper right nest area ($sz_2$), which is also farthest from the mouth zone, showed the lowest amplitudes of these additive motion pulses (Fig 6A).

In the second half of the $nP_d$ 3 session, the amplitudes of the frequency components increased in all surveillance zones, but their peaks were shifted to higher frequencies in the zones in and around the mouth region ($sz_{1,4}$) and remained strongest, and the frequency spectra for the other nest zones at the periphery of the mouth region ($sz_{2-3,5}$) were lower but no longer showed any differences (Fig 6B). This shows that the "gentle motion wave" that initially appeared in the first phase of session $nP_d$ 3 had already subsided in the second phase of this session. By this time, the entire nest had entered MFA, as the basic component of motion activity (blue point sequence in Figs 4–5) also has increased significantly at this time.

## Evaluation of experimental sessions with histogram-based motion spectra

**Semi-Gaussian versus bimodal.** In the present work, histogram-based movement spectra are used. This allows a more detailed description of how the nest evolves from the state of sQu

to the state of MFA in the course of the experimental sessions, and how the MFA intensifies to its peak and then returns to sQu. As shown for episode $mfa_1$ in Fig 7A, the motion spectra of the $nP_d$ phases (blue lines) were monomodal and even Gaussian when the size classes were scaled logarithmically. This motion spectrum under nMFA thus suggests that it is monocausal, i.e. it originates from cohorts at the nest surface that are assigned a uniform task. In $P_d$ phases in nMFA, however, the motion spectra flatten and become bimodal; this is particularly evident in the periphery of the mouth zone ($sz_{1-3,5}$; Fig $7A_{1-3,5}$). This means that under those conditions of the presentation of the dummy wasp, there were at least two cohorts of surface bees, with the larger part on the left side of the motion spectrum remaining quiescent and the smaller part on the right side participating in the collective defense.

**Launching MFA indicated with histogram-based motion spectra.** The MFA phase becomes evident when the monomodal spectra of the corresponding $nP_d$ sessions start to shift towards higher motion magnitudes, and when the bimodal $P_d$ motion spectra in preMFA state become unimodal. For the episode $mfa_1$ this was the case in the both nest areas adjacent to the mouth zone ($sz_{1,5}$) in session $nP_d$ 3, after 2.631 min of experimental time (Fig $7A_{1,5}$).

At this time, in the mouth zone ($sz_4$), the baseline motion level was already much higher from the beginning of the experiment (Figs $4B_4$, $5B_4$ and $7A_4$). With increasing MFA, the $P_d$ spectra showed also here an increasingly weaker superposition of the shimmering components, also with the motion spectra shifting to the higher size classes usually without broadening or flattening the spectra. In the right-side nest periphery ($sz_{2,3}$), the MFA mode started 2 min later, namely in session $P_d$ 4 after 4.4615 min (Fig $7A_{2,3}$), when the onset of MFA showed up in the monomodal shape by a slightly enlarged peak and a slight shift to the right.

During the MFA, the monomodality of the behaviour in the respective cohorts is largely preserved. This becomes visible by the fact that a Gaussian distributed motion pattern then also emerges in the $P_d$ phases when the shimmering has been suppressed. For instance, in episode $mfa_1$ (Fig $7A_{2-3,5}$), the motion patterns in the peripheral nest regions ($sz_{2-3,5}$) remained similar in width, height, and position of their monomodal spectral distribution in sessions $nP_d$ 5 (starting at 5.5832 min) and $P_d$ 6 (at 6.6917 min).

**Fine tuning of the histogram-based motion spectra during MFA.** In principle, the observability of shimmering activities in motion spectra is not limited to bimodal shapes. A quasi-monoidal form of a $P_d$ spectrum in a transition phase from sQu to MFA can exhibit a residual activity of shimmering; this then shows up in a right-dominant skewness of the spectra; one of these cases is displayed in the episode $mfa_1$ in session $P_d$ 4 (Fig $7A_{1-2,5}$). In fact, these residual occurrences of shimmering waves are also visible in the motion profiles (Fig 4A) and in the films (S1 Movie–S4 Movie; for comments see S2 Text) albeit attenuated and only restricted to the right and lower nest regions ($sz_3$). At this time, in session $P_d$ 4 at the experimental time of 4.4615 min, the colony was already in the MFA mode, because the motion spectra on the entire nest were almost or completely monomodal and were shifted to higher motion activities (Fig 7A). This kind of a residual bipolarity still visible in the motion pattern suggests that the front of the mobilization for MFA arrived somewhat later in the peripheral nest regions ($sz_{2-3}$), and thus the capacity for collective defense was still partial at this time and became only later fully suppressed. Based on these topographical distribution patterns of the motion spectra and profiles, it can be assumed that the MFA mode initially started in the region of the mouth zone ($sz_{1,5}$). From here, this mode spreads to the peripheral areas within minutes.

**Recovery of semi-quiescence after MFA.** In the decaying phase of the MFA, the motion spectra shifted again to smaller intensities. Interestingly, this process starts in the left nest regions ($sz_{1,4}$; Fig $6A_{1,4}$) already in session $P_d$ 6 (at 6.6917 min) and intensifies until session $nP_d$ 7 (at 7.8148 min), without changing the spectra in their monomodal form. From then on,

this leftward shift also affected the motion spectra of the regions further to the right of the nest ($sz_{2-3,5}$).

Then, in session $P_d$ 10 (at 11.2183 min), there was for the first time after the peak of MFA, a clear bimodality in the motion spectra in the peripheral right nest regions ($sz_{2-3}$; Fig 7A$_{2,3}$). This means that here in the periphery the state of MFA comes to an end earlier than in the left half of the nest, where it still lasts a few minutes longer ($sz_1$: nP$_d$ 11 / 12.4137 min; in $sz_5$: P$_d$ 12 / 13.5905 min; $sz_4$: P$_d$ 12 / 13.5905 min), and it also mean that defensiveness returns about 2 min earlier in the periphery than in the mouth zone. At this time, the regions near the mouth zone ($sz_{1,5}$) showed still at best implied bimodality in the motion spectra. In the mouth zone ($sz_4$) itself, only a slight, positive skewness can be seen at this time, i.e., a slight preference toward stronger motion components. It is also possible that this is more of a residual restlessness that still stems from the MFA and does not result from the willingness to shimmer. Such restlessness can still be observed here up to session nP$_d$ 15 (Fig 7A$_4$).

## Mass flight mode as characterized by depressions of $P_d$ skewness profiles

The skewness parameter of histogram-based motion spectra (Eq. 6g) describes the equilibrium ($q_s = 1$) or disequilibrium ($q_s \neq 1$) between the components of motion activity during experimental sessions. It is a useful indicator of increases or decreases in the motion profiles, in particular in MFA conditions where motion components of individual cohorts may be strongly masked. It was used for the $P_d$ and nP$_d$ sessions (Figs 7 and 8) to comparatively analyse motion behaviour in the five surveillance zones ($sz_{1-5}$) during the three experimental episodes ($mfa_{1-3}$). Under semi-quiescent, nMFA state, right-sided skewness ($q_s > 1$) associated even with bimodality (Fig 7) is documented under $P_d$ conditions, which is due to the appearance of shimmering waves. As the influence of MFA increases, the motion spectra shift to the right toward higher motion intensities, and at the same time, the skewness in the $P_d$ sessions becomes dominantly left sided ($q_s < 1$), leading to a depression in the time profiles which starts at the onset of the MFA mode and further intensifies until the end of the MFA (Fig 8).

In this context, two depression patterns can be distinguished in the $q_s$ time profiles, and very similar in both episodes $mfa_{1,2}$. In the two peripheral nest regions on the right ($sz_{2,3}$: Fig 8A$_{2-3}$ and 8B$_{2-3}$), the $q_s$ value decreases continuously from the initial phase of MFA to its end. Thereafter, the curve changes quite rapidly to positive $q_s$ values seemingly without any gradation occurring, thus creating a sawtooth shape in the time profile.

In contrast, particularly in the left part of the nest, and here especially in the mouth zone ($sz_4$; Fig 8A$_4$ and 8B$_4$) a double serrated depression of the $q_s$ profile occurs. This phenomenon must be seen, based on two arguments, as a functional property of the MFA process itself: first, it is observable in both independently observed episodes $mfa_{1,2}$ in similar expression; and second, it is shaped by nest-topographic principles as it does not occur exclusively in the mouth zone, to a lesser extent also in the other two observation zones ($sz_{1,5}$: Fig 8A$_{1,5}$ and 8B$_{1,5}$), which are located near the mouth zone, but hardly this double depression occurs in the nest zones peripheral to the mouth zone ($sz_{2,3}$).

## Testing the mass-flight-suspend-defensiveness hypothesis

MFA is not an expression of colony defense but is part of a restructuring of nest functions [42–44]; in this time the colony's ability to defend itself is actually temporarily reduced or blocked as documented by the motion profiles (Fig 4; S1 Movie–S4 Movie; for comments see S2 Text) supporting the *mass-flight-suspend-defensiveness* hypothesis. Probably the most obvious strategy performed by the bees corresponds to the assumption that the ability to defend is reduced proportionally to the amplitude of the motion turbulence occurring during the MFA.

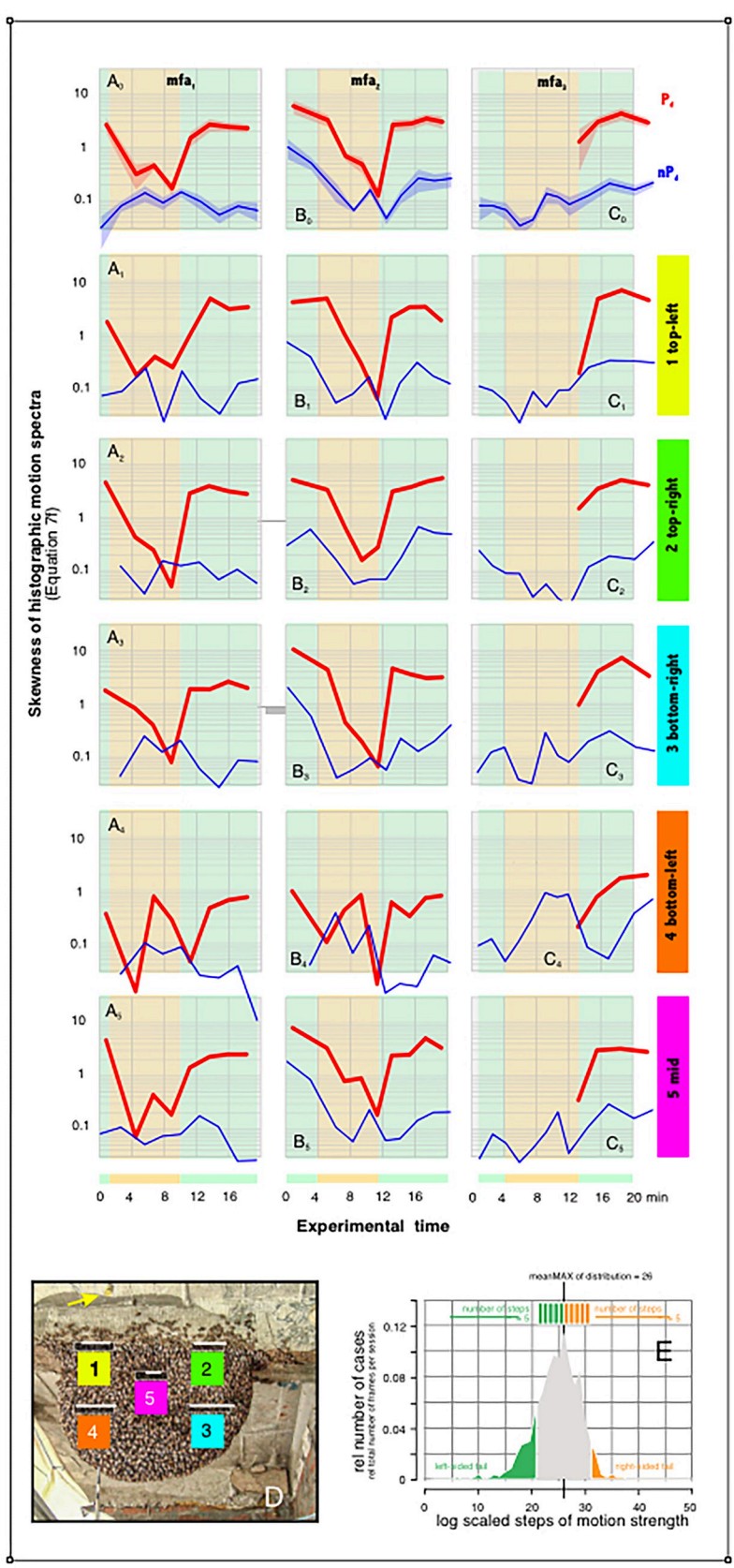

**Fig 8. Skewness of motion profiles in the episodes $mfa_{1-3}$.** A-C, time profiles of the data bodies of three episodes ($mfa_{1-3}$) with respect to the change in skewness of the histogram based spectra in the five surveillance zones $sz_{1-5}$ (see panel D for the map of the five surveillance zones at the experimental nest). Ordinates: the tail area quotient $q_s$ (for definition, see panel E; Eq#6g in S1 Text); abscissa: experimental time in minutes. While episodes $mfa_{1-2}$ alternated between $nP_d$ and $P_d$ sessions, in episode $mfa_3$ the wasp dummy was presented cyclically only after the end of MFA. In order to keep the analysis of episode $mfa_3$ comparable to the other episodes, the corresponding uninterrupted experimental session $nP_d$ 1 ($mfa_3$) for the time before the $P_d$ phases was divided into steps of 5 000 frames. $A_0$-$C_0$, the means (solid lines) and mean errors (transparent clouds above and below the means) of the skewness (Eq#6g in S1 Text) of all surveillance zones $sz_{1-5}$. $A_{1-5}$ - $C_{1-5}$, the skewness (Eq#6g in S1 Text) displayed separately for the five selected surveillance zones $sz_{1-5}$ of the episodes $mfa_{1-3}$; red lines, $P_d$ sessions; blue lines, $nP_d$ sessions. The points of the lines mark the respective sessions ($mfa_1$: with 9 $nP_d$|8 $P_d$ sessions; $n_{ff}$ = 60 713 frames; $mfa_2$: 10|9; 63 340; $mfa_3$: 12 |4; 68 320). D, survey of the surveillance zones $sz_{1-5}$. Orange background of the plot areas denotes MFA, the bright green background denotes nMFA. Panel E, example diagram (session $nP_d$ 1; $mfa_1$; $sz_1$) explaining the definition of left-sided and right-sided tails in the motion spectra with 50 log scaled size classes (Eq#5e in S1 Text). It is the first session in semi-quiescence (nMFA) and without the presentation of the dummy wasp minutes before the colony entered MFA; it is the same data as the first blue point cloud from the left in Fig $4A_1$ and as the first blue-line spectrum in Fig $7A_1$). It covers motion activity over only 0.28 min ($n_{ff}$ = 859) and the sum of the numbers of all partial histographic intervals was normalized to make up the value of 1.0 (Eq#5e in **S1 Text**). The logarithmic scaling of the motion strength levels (Eq#5 in S1 Text) leads to a (semi-) Gaussian distribution of the motion data. For the evaluation of the tail area quotient (Eq#6g in S1 Text) the range of 5 steps between the size class of the referenced maximum of the distribution and the onset of the right-sided ($uT_a$, coded brownish in Panel E) and of the left-sided tails ($lT_a$, coded azure) was chosen. This quotient $q_s$ (Eq#6g in S1 Text) quantifies the relation of both tail areas ($uT_a$ / $lT_a$) and is used as a benchmark for the skewness of a Gaussian distribution of the log motion data. A skewness of $q_s$ = 1.0 signifies a distribution (of log motion strength values) which can be taken as symmetric regarding the magnitude of the left-sided and right-sided tails. At skewness values of $q_s$ < 1.0 the left-sided tail area is larger, which means that the motion strength is developed more at smaller magnitudes; values of $q_s$ > 1.0 denote larger right-sided tail area displaying motion activity with asymmetric prolongation to the larger magnitudes.

This postulates the *inhibition-proportional-to-mass-flight* (*ipm*) hypothesis (Eq#8a-c in S1 Text). The alternative *progressive-inhibition-during-mass-flight* (*pim*) hypothesis (Eq#8d-h in S1 Text) assumes that defensive behavior is progressively inhibited during MFA and that this inhibition is then released in a rapid alternation. The skewness in the motion spectra of the $P_d$ phases (Eq#6g in S1 Text) was introduced in this paper as a parameter to detect the degree of release of shimmering and is used here to test the extent to which the time profile during the $mfa_{1,2}$ episodes agrees with these two models (*ipm*, *pim*). In Fig 9 the empirical data are estimated with the help of the Pearson correlation (Eq#8j in S1 Text) to what extent they fit one of the two models.

Possibly the simplest shape of the skewness profiles is exhibited by those nest regions farthest from the mouth zone (e.g. $mfa_1$: $sz_{2,3}$), as they are typically saw-shaped and decrease steadily during the MFA. Such a decrease of skewness finally also implies the gradual loss of the higher components of the motion activity and can thus be preferentially described by the *pim* hypothesis (Fig 9C and 9D).

However, at the mouth zone ($sz_4$) and nearby ($sz_{1,5}$), the skewness time profiles (Fig 9) show a double dip which are obviously caused by phasic, stronger pulses to inhibit motion activity, which can be roughly associated with the onset and the end of the MFA. This biphasic sink effect is maximally pronounced in the mouth zone in both episodes ($mfa_{1,2}$), and correspondingly less in the two regions adjacent to the mouth zone ($sz_{1,5}$). In addition, episode $mfa_1$ differs slightly from $mfa_2$ in that this double corrugation in skewness time profile is hardly present in $sz_1$, whereas in episode $mfa_2$ this effect is more pronounced in the peripheral $sz_2$ rather than in episode $mfa_1$. Irrespective to such variations between topographical specific courses, the overall time profiles of the skewness $q_s$ show in both episodes ($mfa_{1,2}$) only a slight ripple and small broadening of the sink, and they essentially follow the trajectories of the *pim* hypothesis, rather than the alternative *ipm* hypothesis (Fig 9).

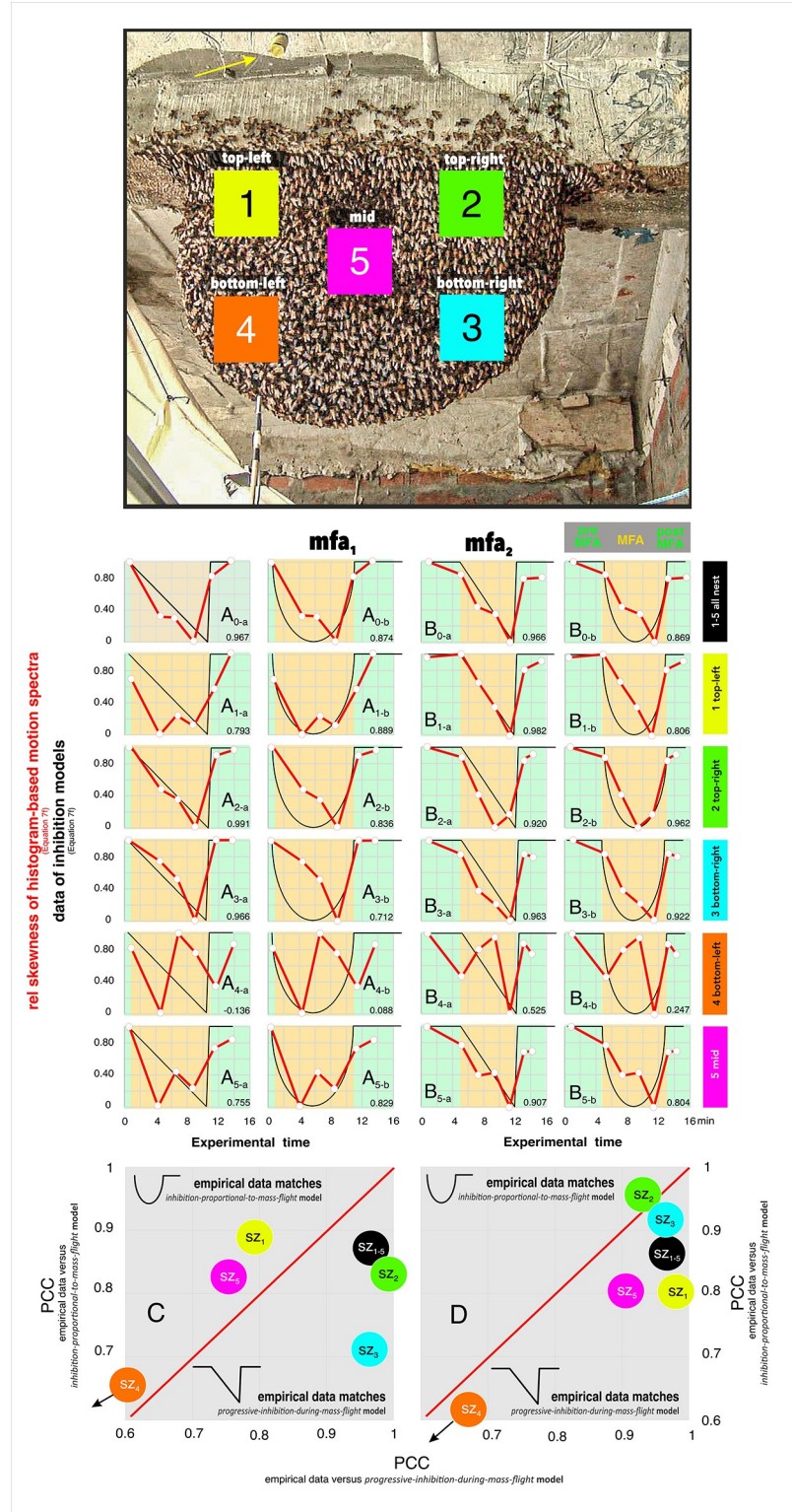

**Fig 9. Skewness of motion profiles in the episodes *mfa₁₋₂*.** Matching the empirical data with two motion inhibition models. The beginning and end of MFA episodes (panel A: *mfa₁*; panel B: *mfa₂*) was determined by manual inspection of the nest and is based on the images taken. The background color green refers to the pre- and post-MFA periods, the background orange to the proper MFA period. Empirical data (red codes in panels A₀₋₅ and B₀₋₅) represent the logarithms of the quotients between the right and left tail activities in the motion spectra of the $P_d$ sessions (Eq#8i in S1

Text). In the *progressive inhibition model* (black lines in the left-sided panel columns of each MFA episode [panels a]; Eq#8a-c in S1 Text), the motion data decreases linearly from the maximum at the beginning of the MFA to the minimum at the end of the MFA. Thereafter, this measure switches back to the maximum. The *proportional inhibition model* (black lines in the right-sided panel columns of each MFA episode [panels b]; Eq#8d-h in S1 Text) includes a semicircular depression from the beginning to the end of the MFA, representing a nonlinear gradual decrease in shimmering activity at the beginning of the MFA and a gradual rebound at the end of the MFA. The empirical data for each of the cases (panels 0: $sz_{1-5}$ panels 1–5: $sz_1$—$sz_5$) as well as the model nomograms are scaled here between 0 and 1. The empirical data are then tested using Pearson's correlation coefficient (PCC) to determine the extent to which they fit either model. The PCC value was noted at the bottom right edge of each panel. Panels C ($mfa_1$) and D ($mfa_2$) summarize these matches for the five surveillance zones (for the color codes of $sz_{1-5}$, see the survey panel on the top and at the descriptions in the side bars of A,B) and for the entire nest (black color code).

## Trade-off between mass flight and defensiveness

In this paper, a suitable pair of parameters was extracted from the histogram-based motion spectra (Figs 7–10D; Eq#9–10 in S1 Text) to quantify the antagonistic relationship between mass flight and defensiveness. The first, abscissa parameter describes the *state of the MFA* (abscissa values in the plots of Fig 10; Eq#10b in S1 Text) by the intensity of collective motion activity in a surveillance zone, with the ability to distinguish the motion state of the MFA from that under sQu. The second, ordinate parameter quantifies the *state of Defensiveness* (ordinate values in the plots Fig 10; Eq#10a in S1 Text) by indicating the extent to which shimmering responses are triggered.

These parameter values are distributed along the nomogram diagonal (Fig 10A and 10B), this correspondence of both parameters in their logarithmised quantities with the paradigm of a classical tradeoff [50, 51, 64] is thus very clear. However, some of these nomographic distributions of the empirical values (Fig 10A$_{1,3,5}$) are flatter and thus miss the direction to the reference coordinate points below right [1,0]. This expresses that, at least on the basis of the metrics chosen here, the defense capability does not drop completely to zero as the MFA heads toward the peak. In addition, all data lie with their centre of gravity shifted into the upper-right area of the nomogram. An adjustment of the trade-off function would therefore be easiest to achieve by modulating the value of coefficient A (Eq#9a in S1 Text). This indicates a somewhat elevated expression of defense, which may be explained by the continuous cyclic presentation of dummy wasp in both episodes $mfa_{1,2}$. Overall, linearity relations can be well documented for the histogram-based (and quantile-based) parameters in the nomograms in all monitoring areas except the mouth area, and their regression polynomials (histogram-based $mfa_1$ [$sz_{1-5}$]: $R^2 = 0.4956$) correspond quite well to the angle of inclination of -45° with the Pearson coefficients (histogram-based $mfa_1$ [$sz_{1-5}$]: PCC = -0.70398; see S1 Table for the data for $mfa_2$). This proves quite well a trade-off effect between MFA and defensiveness.

In the mouth region ($sz_4$), a special case obviously occurs in these nomograms, equally in both episodes $mfa_{1,2}$. The point clouds in Fig 10, like those of the other monitoring areas, are also clearly located in the upper right half of the nomogram, but mostly only a weak linearity can be detected, if at all. Thus, it can be seen that there is no evidence of a trade-off between MFA and defensibility in the mouth area, at least not with these variables of the histogram-based spectra.

In an alternative evaluation, we derived the two parameters (state of defensiveness and of MFA) from quantile-based spectra (according to Eq#10–12 in S1 Text). The results show (quantile-based $mfa_1$ [$sz_{1-5}$]: $R^2 = 0.74189$; Pcc = -0.86133, see S1 Table for the data for $mfa_2$) that for episode $mfa_1$ this approach significantly improves the agreement with the trade-off nomogram, both for the separate consideration of the surveillance zones and overall for the whole nest. In $mfa_2$, there was only a slight improvement for $sz_3$, but a strong improvement for $sz_4$. This shows that the choice of measurement parameters could lead to different weightings

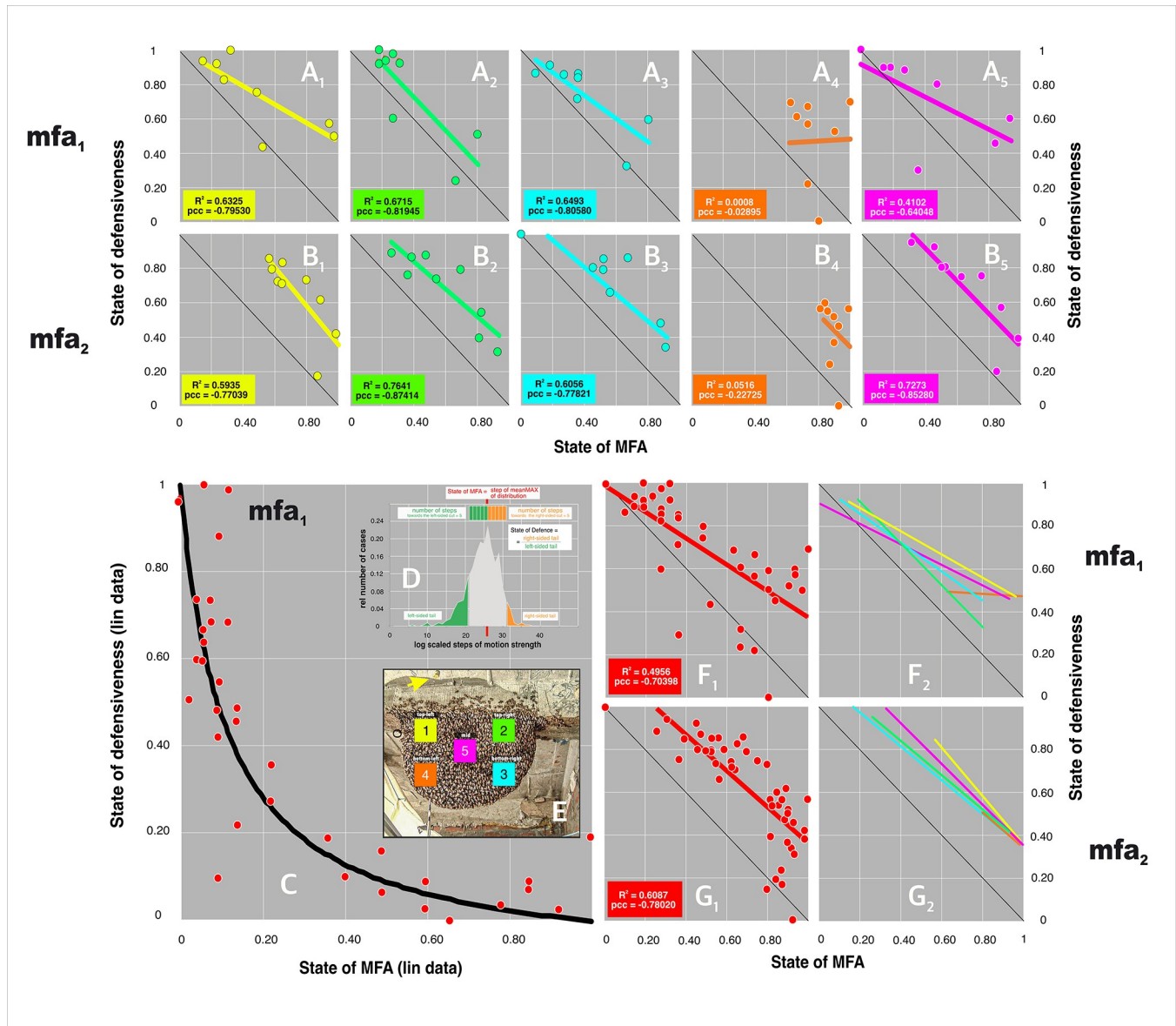

**Fig 10. Tradeoff between the state of mass flight and the state of defensiveness in episodes *mfa₁,₂*.** The nomograms (panels A-C, F-G) describe the relationship between the defense state (ordinate) and the MFA state (abscissa) in five surveillance zones (*sz₁₋₅*; see panel E for color codes) of the experimental nest for two episodes (A: *mfa₁*); B: *mfa₂*). The criteria used for this purpose were selected from the histogram-based spectra and are graphically explained in panel D; abscissa, the state of MFA (parameter $q_s$; Eq#6g in S1 Text); ordinate: the state of defensiveness (parameter k [$P_d$]; Eq#10b in S1 Text). C, the black curve represents the hyperbolic trade-off function (Eq#9a in S1 Text), which serves as a reference for the entire data set (*sz₁₋₅*) of episode *mfa₁,* which were normalized between 0 and 1 and plotted in linear scale. In all other plots (panels A-B,F-G), the empirical data can be compared as double logarithmically scaled with the linear-transformed hyperbolic function (Eq#9b in S1 Text) as the black diagonal line in the range from [0,**1**] to [**1**,0] as reference. Panels A₁₋₅ and B₁₋₅ refer to the data of the surveillance zones *sz₁₋₅* with the their colour codes as displayed in the panel E. The data of panels F₁,G₁ (coded red) display the data of all surveillance zones each and the panels F₂,G₂ summarize the regression lines from the panels A,B in the respective colour codes. Each plot whose coordinates concern histogram-based spectra is additionally described by two key figures: the coefficient of determination R², which evaluates the degree of linearity of the data distribution, and below it the Pearson correlation coefficient, which describes the probability of agreement between the normalised empirical $q_s$ value and the respective y-value (status of defensiveness) of the trade-off model at the corresponding x-value (status of MFA).

for the two trade-off parameters. However, interestingly, the main difference relates exclusively to the mouth zone, where the tradeoff cannot be demonstrated with histogram-based spectra, but can be demonstrated with the quantile-based parameters.

## Discussion

### The phenomenon of MFA in Giant honeybees

An attentive observer will not fail to notice that the swarming during the so-called mass flight of the giant honeybees [35, 42–45, 47] is caused by triggers within the nest and not by external threat. Something similar is known in the western honey bee *Apis mellifera*, there especially this is known as a *cleansing flight* for winter bees [65, 66] or as *Mittagsspiel* for newly hatched bees [67, 68]. At first glance, this restlessness in the nest, as it occurs in MFA, resembles the so-called absconding behaviour of honeybees in general. In giant honeybees, absconding occurs regularly during migration, at comb-right nests or at bivouacs [12, 13]. It is also observed in migratory African [69] and Africanised [70, 71] honey bees, but also generally in European honey bees when they form reproductive swarms or when they migrate in the context of colony collapse disorder [72, 73].

In giant honeybees nests, during MFA state, bees slip through the mesh of the bee curtain from the inside to the outside (see S1 Movie and description at 0.000 min, see S2 Text), crawl around on the nest surface, fly off to circle the nest in a swarm for at least half a minute, and finally land back on the surface layer of the bee curtain. Eventually, the surface layer from the bee curtain may dissolve completely, leaving the bees underneath to form the new surface layer. Thus, with each MFA, the functional architecture of the entire nest is upgraded [42]. How often a giant honey bee colony switches to MFA mode per day depends on the prevailing weather and ambient temperature conditions [42–44], but also on the growth of the colony, which can constantly produce new bees during the reproductive season. Such regular reconstruction of the nest is necessary for the lifestyle of the eusocial bees themselves, which implies that members of different age classes in the nest are assigned new tasks with imperative regularity [18, 68, 74–77]. This includes the exchange of nestmates who form the bee curtain so that they can obtain food and defecate.

Incidentally, a now famous side note is here that *Apis dorsata* causes widespread yellow stains on trees and leaves when swarming during MFA due to defaecation of the bees, and when the nests are near settlements, such stains are regularly observed on laundry hung out to dry. As a consequence of the MFA of giant honeybees, this has always been known to honey hunters and residents living in close partnership with giant honeybees. In the 1980s, at the time of the Cold War, however, this swarming behavior of giant honeybees attracted an unusual amount of military attention and led to a grotesque, historically significant political controversy [78–82], at least for a few years, until these spots, known until then as *yellow rain*, could be scientifically explained as bee poop [80–83].

**Questions to MFA on timing and restructuring of functional nest architecture.**   For the analysis of MFA, two aspects are considered in this paper: First, temporally relevant behavioral markers indicative of self-organization principles of the MFA are identified, particularly with respect to the timing of the onset of the MFA state. This is also partly based on our experience with hundreds of observed MFA episodes [9, 35] in India and Nepal, from which we have clearly deduced that a colony probably determines and decides for itself when it enters this state.

Secondly, the main aspect examined here is the extent to which bees in MFA mode are able to respond with a defensive reaction at all [9, 14, 16, 39, 40]. When an MFA starts up and is in full swing, the nestmates are fully occupied with nest restructuring. This is then also the central statement of the *mass-flight-suspend-defensiveness* hypothesis which is then also subjected to a trade-off analysis.

However, this also raises the question of whether a colony that is unable to develop defensive behaviour is literally blind to external stimuli. Methodologically, this was investigated by

exposing the experimental colony to a cyclically recurring sequence of external threats in the form of the presentation of a dummy wasp (Fig 4).

## The three (a-c) recurrent excitation states of a giant honeybee nest under natural and experimental conditions

In a nest, the semi-quiescence state (a) prevails most of the day, which is more or less the "normal", undisturbed state of life of giant honey bee colonies (S1–S4 movies). In this state, except for the mouth zone, the bees hang in the bee curtain [16, 19, 20, 23] in several layers with the head up and the abdomen down. By external, visual or even mechanical stimuli, the state of shimmering is triggered on demand [9, 27, 36, 39, 40], which is manifested by a wavelike sequence of motion salvos on the nest surface, each involving hundreds of bees simultaneously or sequentially. To an outside observer, such undulating patterns appear like a visible alarm signal that has evolved primarily to deter predatory wasps [9, 40] that would otherwise approach nests unrestricted to strike prey. Furthermore, there is evidence [26, 27] that with this behaviour, the colony is able to inform the nest mates about the current threat situation, even on the other side of the central honeycomb.

This state of semi-quiescence of the nest and its readiness for defensive responses such as shimmering is interrupted at regular intervals during normal daytime operation [42–45] by the state of the MFA (b).

For the present study, the experimental colony was additionally exposed to a recurrent sequence of external stimuli (c) during the sQu and MFA phases, which were themselves autonomous and self-organised [16, 84, 85], alternating stimulation conditions in the form of the presentation of a dummy wasp ($P_d$) with periods of no external threat ($nP_d$). The size, proximity, and speed of this dummy (Fig 1) were adjusted to elicit shimmering behavior under sQu conditions [14, 23, 36, 39]. This type of external experimental stimulation was used here to comparatively measure the ability of cohorts at the nest surface to develop defensive responses also under MFA conditions.

## How a colony regulates mass flight status

The states of sQu (nMFA) and MFA were determined and assessed using parameters of motion activity of individual bees on the nest surface, by parameters of the motion activity of individual bees on the nest surface, calculating *motion profiles* (Figs 4 and 5) and *motion spectra* (Fig 7) on the active side of the experimental nest in five selected surveillance zones ($sz_{1-5}$; Fig 3). These zones differ in their topographical position on the nest and thus also in the behavioral repertoire of the nestmates positioned there.

There are at least two good reasons to assume that the colony commands the start and end of the MFA, with the main decision about the timing of the MFA being made in the mouth zone. In the work, it turns out that the mouth zone, especially during MFA, plays a kind of hub for the entire nest from which self-organised processes [16, 84, 85] are triggered that spread across the nest.

The first argument in favour of these presumptions is based on the observation of characteristic, clearly visible ripples occur in the motion profiles (Fig 6) just before the onset of the MFA, indicating additional agitation limited to one minute or less. This collective incident can be interpreted as a consequence of a signal set by the colony for the start of the MFA mode and can be justified because this phasic motion ripple occurred in the same way in all three episodes ($mfa_{1-3}$) selected for in this work and the topographic differentiation and timing was clearly determined by the proximity of the mouth zone. This underlines the signal character of these ripples and that they originated from the mouth zone and arrived at the other side of the

nest, peripheral to the mouth zone ($sz_{2,3}$), within one minute with decreasing intensity (Fig 6A and 6B).

The second part of arguments relates to the control of how the MFA phase is terminated. Obviously, it seems that the nest can only return to the semi-quiescent state when all bees are back from their defecation flight and a functionally robust surface layer has formed on the nest. The biphasic time course of the spectral skewness revealed a double phasic depression well below the equality value of 1.0 for the $P_d$ sessions, one at the beginning and one at the end of the MFA (Fig 7). It therefore appears that the colony consolidates the termination of the MFA state with a second command. Technically this indicates a *process driven messaging service* (PDMS) [86], which in principle organizes a workflow based on incoming messages about jobs or triggers. Its simplest form would be a non-timed flip-flop switching mechanism [87] in the form of an analog latch that has two data input channels, one of which can be turned on and the other turned off. The so striking biphasic time profile of the skewness data (Figs 8, 9) related to the motion spectra is also nest-topographically determined, with the greatest expression again in the mouth zone and the least expression in the periphery. Furthermore, it is also a strong argument for the reliability of the observation that this double-dip time profile of the skewness data can be similarly demonstrated in the two independently measured MFA episodes ($mfa_{1,2}$) that had the complete program with the dummy's presentation cycles.

### Testing the mass-flight-suspend-defensiveness hypothesis

Any experience with giant honeybee nests in the field very quickly leads to the conviction that MFA [42–44] is not an expression of the colony's defense [14, 16, 24, 27], moreover, that this swarming behavior even blocks the colony's ability to defend itself for a time, supporting the *mass-flight-suspend-defensiveness* hypothesis. Probably the most obvious strategy for how this inhibition proceeds in time is to assume that defensive capabilities are blocked in proportion to the amplitude of the motion turbulence occurring during the MFA which is postulated by the *ipm* hypothesis. This is in contrast to the alternative *pim* hypothesis, which assumes that defensive behavior is progressively inhibited during MFA and that this blockage is then released in a rapid change (Fig 9).

However, it must be stressed here that the question of whether these data correspond to the *ipm* or the *pim* model seems merely academic. It is clear, that statistically, the biphasic sink profiles of e.g. $sz_1$ of the episode $mfa_1$ correspond to the *ipm* model (e.g. Fig 9A$_{1-b}$), and the time profiles determined in the two peripheral areas $sz_{2,3}$ as well as those related to the total nest can be better described by the *pim* model (Fig 9). In reality, this correspondence of the more biphasic sink curves with the *pim* model certainly cannot be justified by the fact that the inhibition of shimmering activity during the MFA is proportional to the general increase in motion activity during the MFA (which is expressed as the blue sequence of dots in the motion profiles of Fig 4). It is much more likely to postulate two independently acting bee cohorts at the nest surface, whose motion activities interfere with each other in the *skewness* profiles. From this it could be plausibly deduced, as noted already earlier, that the mouth zone initiates two control impulses, which have an inhibitory effect on the motion activity, one at the beginning of the MFA and one at its end. The first impulse then sets another slow wave in motion, which spreads from the mouth zone to the peripheral zones of the nest over a few minutes, gradually reinforcing the suppression of the readiness to shimmer and thus also the collective defense. These indicate that defense readiness in MFA mode is generally increasingly inhibited over the course of the MFA episode, but is rapidly reactivated after MFA mode is terminated. And the second inhibition impulse is assumed to trigger the restoration of the bee curtain's defensive capability after the MFA. Obviously, there is an instance in the giant honeybee

collective that gives the command to switch to MFA mode, but also determines its end, i.e. when a new phase of the colony's everyday life can begin, in which the colony returns to a "normal" readiness for collective defense. This seems to work on the principle, already mentioned above, that full defense readiness is only switched on when the MFA phase is actually completed and not before.

## Do colonies perceive external threats in mass flight mode?

This work applies the principle that the behavioural reactions of bees on the surface of an *Apis dorsata* nest also allow conclusions to be drawn as to whether stimuli acting from outside can be perceived by a bee colony or not. For example, the visual pattern of a dummy wasp, as used in the present experiment, regularly triggers in the semi-quiescence state defensive responses in the form of shimmering waves [23, 36, 38, 39]. The motion time profiles (Figs 4 and 5) show that a colony of giant honeybees increasingly reduces its defensive behaviour during the course of MFA activity. This raises the question of whether, after the complete cessation of such shimmering reactions, the bees are still able to perceive external, visual threats at all, and thus may become temporarily blind to nest predators. In the following, we show that the colony's cognitive blindness to external threats does not actually occur during MFA. What is true, however, is that the reactivity with defensive actions is reduced to zero during the MFA, but only in a small time range and then not affecting the entire nest.

One indication of threat perception across MFA is that external threats amplify the scale of mass flight. The observations in the motion profiles (Figs 4 and 5) show in three independent episodes that a cyclically recurring external threat, as carried out in the experiments, causes a colony to significantly increase recruitment for MFA. Therefore, it can be assumed that the much stronger basal motion in the MFA during the sustained cyclic presentation of the dummy wasp (Figs 4AB and 5AB in contrast to Figs 4C and 5C) can be taken as evidence that the bees remaining at the nest do not completely switch off their ability to perceive external threats when their nest mates are at defecation flight and the functional nest architecture is quite disrupted. This is surprising as the nest shows no reactivity in the form of shimmering during a substantial part of the MFA. Although these results are based on only three MFA episodes and thus represent basically only a single case study, this finding is stringent precisely because the results refer to one and the same experimental colony.

Another indication is that kurtosis of histogram-based spectra traces cohorts responding to external threats. The bimodal histogram-based $P_d$ spectra with a right-sided skewness ($q_s > 1$; Eq#6g in S1 Text) characteristic of the nMFA state become unimodal as soon the MFA mode inhibits shimmering (Figs 7 and 8) and, moreover, they may then also show a left-skewed kurtosis ($q_s < 1$). This indicates a residual bimodality, which then also leads to strong dips in the time profile of the skewness. These are detectable in both episodes $mfa_{1,2}$, progressing gradually until the end of the MFA phase, and are particularly visible in the peripheral nest regions $sz_{2,3}$ (Fig 8A$_2$ and 8B$_2$).

This left-sided kurtosis in the $P_d$ spectra may be caused by two factors: Firstly, in MFA mode, this could emanate from a smaller group of agents in the corresponding surveillance zone, which react to the presented wasp dummy with somewhat stronger motions than probably most of the others, but without showing abdominal flipping as in shimmering. Their share becomes noticeable as a steeper right flank of the $P_d$ spectra at the end of the MFA (Fig 7A$_3$ for sessions $P_d$ 6–8; Fig 7A$_4$ for session $P_d$ 10). A second, additional possibility leads to a left-sided kurtosis when the motion activity of reactive agents is inhibited by the presence of the wasp dummy. This could explain the broader band of motion strength on the left tail of the motion spectra, which becomes somewhat wider towards the end of the MFA (e.g., Fig 7A$_4$ in the

sessions $P_d$ 12–14, and Fig 7A$_2$ in the sessions $P_d$ 8–10). In either case, this would mean that both cohorts remain susceptible to external threats even in MFA mode, but behave in opposite ways. Neither, however, would be capable of a defensive response.

## How to keep risk low while expanding collective nest functions?

This raises the question of what dangers a bee colony exposes itself to when it switches to MFA mode. The fact that such MFA management tasks in the nest have to be addressed regularly, sometimes in 5-hour increments, shows how vital they are for the development of a colony. It can be assumed that their execution and performance has been designed in the course of evolution to ensure the survival of the colony and thus of the entire species in the corresponding habitat. In giant honeybees, a number of factors (a-f) can be listed as the basis for a colony's ability to cope with the necessary nest restructuring during such an MFA phase without repeatedly exposing themselves to mortal danger through the temporary loss of defense capability.

**Tradeoff between defensiveness and MFA (a).** Trade-offs arise from constraints of many origins [50], including simple physics [88, 89], in terms of labour or economy [90, 91], and are usually made in social communities by genetically controlled principles of self-organization [16, 84, 85]. In these concepts, time management is always an important factor that plays a special role when several tasks should be completed at the same time. Obviously, the bee colony must tactically weigh a list of advantages and disadvantages, and also find a number of short-term organizational solutions to reduce or even completely abandon important collective services for a period of time. This includes foraging, which seems to be less reduced during the MFA phase. At least, waggle dancers are consistently observed in the mouth zone (S1 Movie– S4 Movie; for comments see S2 Text) also during MFA, which means that workers continuously return from their foraging flight. This is plausible for two reasons; first, such workers have departed for the collective flight even before the nest is fully switched to mass flight mode; and second, during MFA, the mouth zone is kept autonomous relative to the rest of the nest regions. On the other hand, the ability to defend itself is in Giant honeybee nests completely suspended for a some minutes. Finally, defence readiness begins when all nest members are ready and able to do so again, or at least with a strategically selected group that keeps its eyes open to detect threats from outside. In such a case, those involved in the defence must then also be able to alert the nest, recruit soldiers and possibly release them. So when a giant honey bee colony decides to go into MFA mode, it accepts that this will throw the nest structure pretty much out of whack. The results show that most regions of a giant honeybee nest cope quite well with such a trade-off situation (Fig 10). Interestingly, the mouth zone is a kind of exception here (Fig 10A$_4$ and 10B$_4$), less concerned with handling the trade-off between MFA and defensiveness. To a certain extent, it retains its function as a hub for various important tasks in the nest, such as foraging, guarding, but also in this respect to signal the start and end of the mass flight.

**Time scheduling of MFA (b).** The factor time refers here to how long an MFA should or must take to make an appropriate adjustment to the functional nest architecture. This is especially necessary due to the cyclic appearance of new bee generations in order to delegate new tasks to the cohorts in the nest [75, 77]. In the course of evolution, this duration has probably turned out to be the indispensable brevity of such a phase with the rule, the shorter the remodeling in the nest lasts, the lower the chance of potential nest predators to attack a defenseless colony. The observations show (Figs 4 and 5 and 7–9) that the average time during which a colony becomes effectively defenseless and functionally blind to nest predators is, after all, limited to only a few minutes.

**Graduality in MFA (c).** The same results also show that the critical situation a colony finds itself in because it is helpless against external predators is mitigated by the fact that the transition into MFA mode is not abrupt, but gradual over a few minutes. The mouth zone, as an integrative interface in the nest, is predestined to control cross-nest behaviours such as MFA. Apparently, the signals for the transition into the MFA mode spreads from here to the adjacent nest regions. It follows that the entry into the MFA mode in the periphery and thus also the inhibition of defensive readiness begins later there than in the mouth zone and then also ends earlier, which also means a gradual and not abrupt loss of defensive readiness for the colony as a whole.

**Rapid switch to semi-quiescence (d).** The transition from MFA to "normal" mode, however, then takes place in *a rapid switch-on* (d) *of semi-quiescence* (Figs 8 and 9) with the full capacity of defense. The entire transformation of nest functions is far from complete when the nest appears to have returned to semi-quiescence and can provide full defence. This is shown by results of measuring the cover thickness of the bee curtain [89]. This means that some communal tasks, such as defensive readiness and, to a lesser extent, foraging, are only disrupted for a relatively short time by the interruption of the MFA mode. Many other activities that serve to maintain internal nest structures, such as nest building, larval care, egg laying by the queen, guarding or trophallaxis, can be so profoundly affected by MFA that it takes at least another 10–20 min longer for these activities to return to normal.

**In nest aggregations, MFA episodes invariably occur in strict succession (e).** Even if MFA episodes occur periodically and thus regularly, they cannot be predicted exactly to the point in time. This is not primarily determined by the ambient temperature [43, 44], but it remains the autonomous decision of each colony for itself to determine the actual start time for the MFA mode. The detection of a start signal (Fig 6) and a putative PDMS [86] decision mode demonstrates this self-organized, autonomous control of MFA timing by the colony. This also becomes particularly clear when individual colonies in a nesting aggregation of ten or even a hundred colonies switch to MFA mode. Interestingly, this does not happen by chance, because then one would expect that at least several colonies in such an aggregation would also swarm simultaneously. According to our experience with hundreds of observed MFA episodes in colonies of *A. dorsata* [9, 35, 48] and *A. laboriosa* [49], the MFA episodes invariably occurs in strict succession, with one nest after another entering MFA mode. However, such a series of MFA in an aggregation of nests already takes place in a predictable time window of several hours, for example in the morning from 10 to 12 am and in the afternoon from 12 to 14 pm [43, 44]. A possible reason could also be that this time scheduling minimises the drifting behaviour between neighbouring colonies [92].

**Swarming in honeybees mostly signals peril for nest predators (f).** The swarming of a large part of the colony during MFA in an area of about ten metres around the nest probably triggers an ambivalent message to nest predators. On the one hand, swarming in general can signal danger, because bee swarms may well consist of guard or soldier bees that are ready to sting, ready to fight and thus also highly reactive. Such swarming bees are often recruited specifically for an impending emergency, prompting them to pursue the potential predators to drive them away [9, 24, 27]. With the strictness of the sequence of MFA swarm activity in a colony aggregation, a kind of double benefit for the neighbouring colonies can be seen: Colonies that are currently in the vicinity of colonies that are in MFA mode can gain some territorial defence protection through the repulsive signal of swarming itself. Conversely, the neighbouring colonies that are not in MFA mode are fully defensible and can also extend this defensive protection against potential nest predators to the colonies in MFA mode by the mere neighbourhood effect.

On the other side, swarming itself has a counterproductive property, because it informs potential predators of the presence of the nest. Bee-eaters or honey buzzards [9, 24, 27] gain rapidly knowledge through scouting experience where the nests of giant honeybees are distributed in their area. Nevertheless they perceive such swarm-active nests of giant honeybees already at a distance of several hundred meters and can thus also locate a newly arrived bee colony. Moreover, such an MFA episode may not be considered dangerous to them by the predators, at all if they are MFA episodes with drones. These occur at particular times of the year, preferably at dusk, and may be the target of hunting bats precisely because of this particular time of day (own observations at JNU in Delhi [45, 46]). In doing so, both sides have obviously tried to adapt, on the one hand the bees, which even move such an MFA to dusk in order to be less conspicuous, and on the other hand the inventive bats, which start their daily predatory flights exactly then.

## Supporting information

**S1 Text. Equations.**
(DOCX)

**S2 Text. Comments to the movies.**
(DOCX)

**S1 Table. Comparison of the empirical data of the defensiveness and the MFA state in Giant honeybees with the theoretical tradeoff function.**
(PDF)

**S1 Movie. Movie documentation of the mass flight episode *mfa$_1$* recorded from the experimental nest concerning motion activity in the surveillance zone *sz$_1$* ranging from start of nP$_d$1 to end of P$_d$4 (from frame 1 [4.9470 min experimental time] to frame 32 177 [10.7270 min]).**
(MP4)

**S2 Movie. Movie documentation of the mass flight episode *mfa$_1$* recorded from the experimental nest concerning motion activity in the surveillance zone *sz$_1$* ranging from start of nP$_d$5 to end of nP$_d$9 (from frame 14 876 [4.9470 min experimental time] to frame 14 876 [4.9470 min]).**
(MP4)

**S3 Movie. Movie documentation of the mass flight episode *mfa$_1$* recorded from the experimental nest concerning motion activity in the surveillance zone *sz$_1$* ranging from start of P$_d$10 to end of nP$_d$13 (from frame 32 177 [10.7270 min experimental time] to frame 46 463 [15.4897 min]).**
(MP4)

**S4 Movie. Movie documentation of the mass flight episode *mfa$_1$* recorded from the experimental nest concerning motion activity in the surveillance zone *sz$_1$* ranging from start of nP$_d$14 to end of nP$_d$17 (from frame 46 463 [15.48967 min experimental time] to frame 60 704 [20.2373 min]).**
(MP4)

## Acknowledgments

My collaborators on site: My Master's student at the time, Dominique Waddoup, and my PhD student at the time, Frank Weihmann, helped set up the experimental facility and collect data

at the locations in Chitwan (Nepal). We thank Dr. Madhu Singh, Dr. S.M. Man, Dr. R. Thapa and Dr. M. B. Gewali from Tribhuvan University, Kathmandu, Nepal, for their support regarding logistics, and Reinhold Stachl (NPN electronics) for providing infrared cameras for several expeditions to India and Nepal. We thank the University of Graz for covering the publication costs.

## Author Contributions

**Conceptualization:** Gerald Kastberger.

**Data curation:** Gerald Kastberger, Martin Ebner, Thomas Hötzl.

**Formal analysis:** Gerald Kastberger.

**Funding acquisition:** Gerald Kastberger.

**Investigation:** Gerald Kastberger, Martin Ebner, Thomas Hötzl.

**Methodology:** Gerald Kastberger, Martin Ebner, Thomas Hötzl.

**Project administration:** Gerald Kastberger.

**Resources:** Gerald Kastberger.

**Software:** Gerald Kastberger, Martin Ebner, Thomas Hötzl.

**Supervision:** Gerald Kastberger.

**Validation:** Gerald Kastberger.

**Visualization:** Gerald Kastberger.

**Writing – original draft:** Gerald Kastberger, Martin Ebner.

**Writing – review & editing:** Gerald Kastberger.

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
