## [Decision Letter · Decision Letter 0]

19 Jul 2023

PONE-D-23-15840Giant honeybees (Apis dorsata) trade off defensiveness against periodic mass flight activityPLOS ONE

Dear Dr. Kastberger,

Thank you for submitting your manuscript to PLOS ONE. After careful consideration, we feel that it has merit but does not fully meet PLOS ONE’s publication criteria as it currently stands. Therefore, we invite you to submit a revised version of the manuscript that addresses the points raised during the review process.

In the opinion of the Editor and Reviewer #1, the manuscript should be shortened significantly. See Reviewer #1's report for a number of constructive suggestions. Please also address the point of criticism regarding the small sample size (three videos from one nest). Reviewer #2's criticisms are more serious. In her/his opinion, the methodology (dummy wasp) is suboptimal. In particular, Reviewer #2 points out that honey bees also respond to olfactory cues from predators. Please justify your experimental design or adjust the conclusions if necessary.

We look forward to receiving your revised manuscript.

Kind regards,

Wolfgang Blenau

Academic Editor

PLOS ONE

The name of the colleague or the details of the professional service that edited your manuscrip

“Funding: This work was supported by the Austrian Science Foundation (FWF Project P 20515-B16) URL: https://www.fwf.ac.at/en/. The funders had no role in the study design, data collection and analysis, decision to publish or preparation of the manuscript.

The authors gratefully acknowledge the financial support for the publication costs by the University of Graz”

5. Please amend your authorship list in your manuscript file to include authors Martin Ebner and Thomas Hötzl.

6. We notice that your supplementary figures are uploaded with the file type 'Figure'. Please amend the file type to 'Supporting Information'. Please ensure that each Supporting Information file has a legend listed in the manuscript after the references list.

7. We note that Figures 1 and 2 in your submission contain copyrighted images. All PLOS content is published under the Creative Commons Attribution License (CC BY 4.0), which means that the manuscript, images, and Supporting Information files will be freely available online, and any third party is permitted to access, download, copy, distribute, and use these materials in any way, even commercially, with proper attribution. For more information, see our copyright guidelines: http://journals.plos.org/plosone/s/licenses-and-copyright.

1. You may seek permission from the original copyright holder of Figures 1 and 2 to publish the content specifically under the CC BY 4.0 license.

Reviewers' comments:

Reviewer's Responses to Questions

**Comments to the Author**

1. Is the manuscript technically sound, and do the data support the conclusions?

Reviewer #1: Partly

Reviewer #2: No

2. Has the statistical analysis been performed appropriately and rigorously? 

Reviewer #1: I Don't Know

Reviewer #2: I Don't Know

3. Have the authors made all data underlying the findings in their manuscript fully available?

Reviewer #1: No

Reviewer #2: No

4. Is the manuscript presented in an intelligible fashion and written in standard English?

Reviewer #1: No

Reviewer #2: Yes

5. Review Comments to the Author

Reviewer #1: FORM:

Only 1 author is listed in the submission system, but there are 3 on the manuscript? Is that a mistake?

While I appreciate that Plos ONE has no length constraints, this paper is just way too long. I think that it could be re-structured to a much clearer and shorter format without losing information. If I take the introduction as an example, at least 3 paragraphs (L48-60, 112-128) could be deleted without any inconvenience for the reader. Maybe the most striking example of this lack of conciseness is the fact that even the historical anecdote about the yellow rain, which is an interesting piece of trivia but far from being relevant to this paper, is narrated twice (in the introduction and again in the discussion).

In addition, the structure is all over the place. The results are sorted according to the analysis method instead of the biological finding, resulting in lots of repeats and circling back. The discussion starts with what is basically a 2nd introduction and then includes a mix of old and new results, with p values that suddenly appear for the 1st time at this point, before finally turning into a proper discussion. Some of the main figures are barely used (e.g. Fig 12 is only cited in the methods and discussion), whereas Suppl. Fig 1 seems to correspond to an entire section of results. Fig. 3 (cited >70 times!) has some panels explained at the beginning, and some much later (after Fig. 5), so it’s not very clear why they are grouped on the same figure. These may sound like petty considerations, but they accumulate into a very confusing and hard to read paper.

As a final point regarding the form, the vocabulary employed throughout the paper is extremely mathematical. I will admit to not being mathematically inclined and I’m not in favour of “dumbing down” anything, but I think that many of the parameters could be named and presented in ways that are more intuitive to people like me (i.e. biologists who do not handle these terms regularly, which will probably be your main audience). For example, it took me some time to grasp that “differential quantile motion activity” is basically a measure of variance.

CONTENT:

Through all this circumvoluted analysis of the data, what I take from this paper can be summarized quite simply:

-there may be a tremor that signals the start of the MFA

-the MFA spreads forth from and back to the mouth zone

-Bees don’t shimmer during the MFA (thus trading off between defence and other nest functions), but might instead freeze in response to threats

-Loss of shimmering is progressive but its return is abrupt, and occurs before the bees have fully settled back

These conclusions are interesting and well-supported by the data, but the sample size is very small (3 videos analyzed, all from the same nest). How were these videos selected out of the ~80 that had apparently been recorded? Why use only a single nest?

Abstract:

Overall not very understandable, because too many terms are introduced with implicit definitions (Pd, nPd, episode, session, the saw-tooth like shape of the motion spectra…). It makes sense now after reading the paper, but it didn’t before.

L57-58, “Their way of life includes the ability to collectively exploit niches of food sources that may even be spurned by other flower-visiting species”. What do you mean by that? Is there a paper that could be cited?

L111: speculative

L150: Souldn’t it be “nPd sessions where no dummy was presented”?

L215: preferably? It faces the sun or it doesn’t

L248-258: I didn’t understand the point of this paragraph. It sounds like luminance variations due to the weather are acknowledged but ignored? Significant changes might not occur between 2 frames, but they could between sessions

L268: What are the erosion and dilation filters doing?

L318-322: This may be my own limitation, but it did not understand what a quantile based motion spectra is from this explanation

L378-392: I was left with the feeling that Qmot is not a great parameter to look at, since its interpretation is often limited

L583, 1164 and in other places: Concluding that the wasp presentations increase the MFA amplitude based on a single counter example is very far-fetched. One would need, at the very least, another example of a case without stimulations.

L588: Fig 5 does not have a panel D, nor E

L602: specified?

L609: The low frequency component still has the highest weight in most zones, as far as I can see. That contradicts your interpretation

L639-644: it’s really hard to see onset times on this figure, and the yellow colour on the vertical axes doesn’t match the start and finish of the MFA as described here.

L701: “but may even have been inhibited by the external threat scenario at work during this period.” How do you reach this conclusion?

L736-756: I didn’t understand the point of this section. 2/3 of it defines qs, which is then not used.

L773: “positive qs” I thought that qs was compared to 1, not 0?

L778-782: I get why the start and finish of the MFA are important time points, but how does that explain the “middle” peak in between the dips?

L801 and in other places: Does the word “significant” here actually refers to a statistical test? If so, where is this data?

L808-810: “some cohorts are frozen…”. Where am I supposed to see that? The red curves don’t seem to go much below the blue ones, and they wiggle up and down during the MFA.

L850-851: I don’t see that, the red curves mainly go right rather than up

L919: where is this line of equality?

L943: Fig 11B doesn’t have a time axis?

L1143-1146: This PDMS thing comes out of the blue and doesn’t seem necessary to explain that there are a switch on and a switch off.

L1204: this slope parameter seems overly complicated. Wouldn’t the variance or the difference max-min give a similar, more straightforward measure?

L1222 and in other places: Concluding that the colony “cannot defend itself” is too broad. They can’t shimmer, but we don’t know anything about stinging or other forms of defence.

L1234-1240: Why are these models appearing only now? What are they exactly? They are not even described in the methods!

L1241: “there is an instance that gives the command” very speculative

L1277: there is no proof that bees have an assigned place

L1344-1347: What does it means if the quantiles match better than the histogram?

L1464 and in other places: “Some cohorts are quiescent while some aren’t”. I’m not sure that this conclusion can be drawn from an analysis at the pixel level. Some pixels on the bee likely remain the same even when the bee is moving, and conversely pixels at the edges of bees are more likely to vary. This noise could be fairly constant and make up the “quiescent” peak.

Figures:

Fig. 3 GH: it would be nice to have Q=1 highlighted.

Fig 7: All other figures just plot the 5 zones separately, why is this one plotting weird combinations of zones??

Fig 8: not same colour code between panels (orange/yellow etc).

Why are episodes 3a-3b on Fig 9 becoming 3-4 on Fig 10? Please keep the numbering constant!

Fig 11 and S4 are exactly the same?

Fig 11E should be annotated with the episode numbers

Reviewer #2: It is known that more than half A. dorsata colonies could perform daily periodic mass flights ranging in number from zero to six. This behaviour apparently play a role in cleansing and orientation flights of young worker bees. However, the role of MFA in

nesting biology,behaviour as well as its ecological significance remains unclear. Therefore it is important to learn more about this phenomenon. This study investigate how a colony in MFA be mode while reconfigure its internal organization which can maintain its defense capability. The authors using a dummy wasp to elicit shimmering behavior on the nest surface to measure whether external threats can be perceived collectively in MFA mode. From my personal perspective, this method is not advisable, I believe a dummy wasp visually could induce a wavy shimmering pattern, but there are many defensive ways in honey bees, for example, bee alarm pheromones and other physical alarm signals may be involved, honey bees detect predators also dependent on chemical signals from the predator at the same time. So when a colony in MFA mode, the flying bees might ignore visual stimuli, but it is possible that they increase their chemical defence, so it doesn't mean that the collective defense capability of the colony is undermined. On the contrary, such a state of social arousal is triggered preferably by external visual stimuli, usually leads to a synchronized, collectively well-organized arousal of many bees at the nest surface and even drives the wasps out of the area of the bee nest.

Furthermore, There are few mistakes in the manuscript need to be revise.

Line 54 Serial number of the reference is [2, 9-14], but in the Line 65 the Serial number of the reference is 23,27 and 44, please check it.

6. PLOS authors have the option to publish the peer review history of their article (what does this mean?). If published, this will include your full peer review and any attached files.

Reviewer #1: No

Reviewer #2: No

---

## [Author Response · Author response to Decision Letter 0]

24 Oct 2023

Rebuttal letter / Notes for the editor / Response to Reviewers

Notes for the editor:

• The paper was completely revised, with texts and illustrations renewed from scratch. The editing itself happened in several, at least 10 rounds. Therefore, it does not make any sense to submit a marked-up copy of the manuscript that highlights changes. Sorry for this.

• Most of the repetitions were taken out and only left in individual cases when the readability of the text had to be supported. Several contents in the Methods and Results section (e.g. quantile-based spectra analysis, coverage of the thickness of the bee curtain) were totally removed in order to shorten the paper considerably.

• In the chapter Methods I have given the hint that the abbreviation "Eq #" refers to the respective Equations in the corresponding file (S1 File Equations). I think that this makes the text much easier to read, but I may have to change this rule at the request of the editors.

• The abbreviations of mass flight episodes (mfa) and surveillance zones (sz) are in italics, those of “mass flight status” (MFA) and experimental sessions (Pd and nPd) are in normal format. Italic are kept for all variable names in Equations. This, however, refers mainly to the supplementary Equation File.

• In this version of the manuscript, I used the following abbreviation for the citing figures: e.g. “Figs 4,5” or “Fig 4A4,B4” or “Fig 4 and S1-4 Movies”. Other formatting is easily to be organized. 

• Ad your query in your email: “We note that Figures 1 and 2 in your submission contain copyrighted images. We require you to either (1) present written permission from the copyright holder to publish these figures specifically under the CC BY 4.0 license” . I have noted in the capture of Figs 1,2 and of the S1-4 Movies the statement “Copyright holder is the author (GK).” If this is not sufficient you are free to alter the statement accordingly.

 

Response to Reviewers

Thank you for the kind comments of the editor and the two reviewers. In summary, the comments have resulted in a much more stringent, non-repetitive text, and have contributed to a significant improvement in quality. The paper now focuses on the main topic of the paper, namely the trade-off between updating the functional architecture in the giant honeybee collective and its ability to defend itself against nest predators.

In the following, I will answer most of the reviewers' questions.

Reviewer #1:

“While I appreciate that Plos ONE has no length constraints, this paper is just way too long…”

My comments: 

• Many of Reviewer #1's questions are now redundant and no longer relevant, as the text of the paper has not only been changed from the ground up, but also significantly shortened. The paper has been completely revised, with texts and illustrations renewed from scratch. The revision itself took place in several rounds, so it makes no sense to show the original text with its revisions.

• Most of the repetitions were taken out and only left in isolated cases, but only when the readability of the text needed to be supported. Several contents in the Methods and Results section (e.g. quantile-based spectra analysis, recording the coverage of the bee curtain) were completely removed in order to shorten the paper considerably. Parts of it will be published in a separate further paper.

Reviewer #1:

“The results are sorted according to the analysis method instead of the biological finding, resulting in lots of repeats and circling back. “

My comments: 

• The sequence of subchapters in Results first takes care of how mass flight (MFA) episodes are observable. Here, command signals for the start of MFA are found in the time profiles of motion activities. The method of histogram-based motion spectra, which allow monitoring the fine structure of collective motion activity before, during and after mass flight, is then developed. With these tools, it is then also possible to prove the mass-flight-suspend-defensiveness a hypothesis and, furthermore, the trading-off effect between defensiveness and MFA.

• I think that this sequence of chapters is now harmonised from the simple basics of measurability to the development of complex relationships. It is clear that the methods section follows this sequence closely to provide all the definitions and tools in this sequence. The discussion, however, then summarises the behavioural topics, although even here the primary sequence adopted is essentially adhered to: from the description of MFA to the testing of the hypothesis of mass flight defence, with the related question "Do colonies perceive external threats in mass flight mode?" to the summary of "How can a colony keep the risk from highly adapted predators low while expanding collective nest functions?". Here, I derived at least four aspects from this measurement and observation programme of this present paper: the tradeoff between MFA and defensiveness, the timing and gradualness of MFA at the nest, and the rapid switch to semi-quiescence after MFA. Two further aspects (desynchrony of MFA episodes in nest aggregations; signalling of danger of swarms of stinging honey bees to potential predators) relate to our long experience with MFA in nest aggregations and with interactions with nest predators in giant honey bees in general.

• I admit that there are quite a few jump-backs still in the revised text, but this is due to the intention that each chapter, such as the discussion or the introduction, should also be self-contained in a certain way in order to achieve a nevertheless complete framework of understanding without the need for cross-reading.

Reviewer #1:

“ I will admit to not being mathematically inclined and I’m not in favour of “dumbing down” anything, but I think that many of the parameters could be named and presented in ways that are more intuitive to people like me …”

My comments: 

• I have now, after these sensible remarks by Reviewer #1 deliberately avoided writing "mathematising" texts. The consequence of this is that equations and algorithms are now dealt with exclusively in the Methods section or in a Supplementary material file S1 File Equations. Reading the main text, especially the Introduction, Results and Discussion (as well of course the Abstracts and the captions) is now, hopefully, much easier. 

• Although the text has been toned down by omitting mathematised passages, it was of course still necessary to include descriptions of images, time profiles and motion spectra in the Results section. I admit that this might make the reading a bit dry, but in all cases this happens I always tried to summarise the related aspects afterwards with simpler formulations.

• The variables derived to represent the phenomena were described in all stages of their algorithm in the supplementary material.

• In the Methods chapter, I have thereby given the hint that the abbreviation "Eq #" refers to the respective Equations in the corresponding file (S1 File Equations). I think that this makes the text much easier to read. If the editors of PloS ONE so wish, I will of course change this rule I have established.

• The additional material is now essentially limited in a first part to the derivation of variables. 

• A second block (S1-4 Movies) gives the examples of the original film sequences describing the behaviour of the experimental nest during the entire recorded episode mfa1 which happens in a period of about 20 min. The motion activity data monitored for this purpose in the images only refer to the surveillance zone of sz1. 

• The pixel resolution of these films is such that this episode was divided into four partial films; the volumes of these films correspond to the publication rule of < 15 M each. If the editors would like higher quality copies, I can provide them. 

• These films are particularly valuable as an appendix to this publication also for this reason, because they get to the heart of what this paper is saying well, at least with one episode and the data from a selected surveillance zone at the nest. 

• Thus, when looking at the film and the displayed data, it may be easier to understand the difficulties encountered in obtaining the metrics behind.

Reviewer #1:

“positive qs” I thought that qs was compared to 1, not 0?”

My comment: 

Thank you for pointing this out, unfortunately the mistake happened because additive relativities were also used as difference analysis in the original manuscript. In the revised manuscript I correctly speak of left-sided or right-sided skewness of the (histogram-based) spectrum. (Several others of Reviewer 1's questions have now also become superfluous due to the completely revised text. These are no longer addressed in this Rebuttal letter.)

Reviewer #1:

“What are the erosion and dilation filters doing?”

My comment: 

These are common digital filter functions of image analysis that do not need to be explained further in a paper like this.

Reviewer #1:

“Abstract:

Overall not very understandable, because too many terms are introduced with implicit definitions (Pd, nPd, episode, session, the saw-tooth like shape of the motion spectra…). It makes sense now after reading the paper, but it didn’t before.”

My comment: 

Now I have redesigned the Abstract so that it can be read before the main text of the paper.

Reviewer #1:

“: I didn’t understand the point of this paragraph. It sounds like luminance variations due to the weather are acknowledged but ignored? Significant changes might not occur between 2 frames, but they could between sessions”

My comment: 

The key point here is that motion activity is assessed pixel by pixel by different luminance values between two successive images. The speed at which the luminance values per pixel area change is very slow due to environmental conditions (the fastest event here could be when a cloud edge starts to block the sun), but it is much faster when the reflective surfaces of a bee (such as wings, legs or head, thorax or metasoma) are moved in one direction or somehow rotated. As a result, luminance values can change within milliseconds. Keeping this in mind, the above criticism is not coherent for me.

Reviewer #1:

“Some cohorts are quiescent while some aren’t”. I’m not sure that this conclusion can be drawn from an analysis at the pixel level. Some pixels on the bee likely remain the same even when the bee is moving, and conversely pixels at the edges of bees are more likely to vary. This noise could be fairly constant and make up the “quiescent” peak.”

My comment: 

That is a good critical approach. I still think this is a still valid textual approach to the form of biphasicity of motion spectra that I have chosen. The point is that although we are measuring at the pixel level, the result is a probability measure of the motion activity of the entire surveillance zone at the corresponding time of the frame. Assuming that the histogram-based spectrum is biphasic, this can mean the following: This is then a Pd session in semi-quiescence (before or after the MFA), and this lasts 1 minute, so we get 60s * 50 frames = 3000 observations at even so many time points. Of these, let's say 600 frames are associated with the higher active peak in the spectrum of this experimental Pd session. Since the shimmering rate over the whole nest surface is about 1 Hz, we do not get all of them by tracking, but perhaps many of them also in the respective surveillance zone. We know from other research (Kastberger et al 2012, “How to join a wave”; or Kastberger et al 2014, Speeding up social waves) that there are several cases of how surface bees can actually join a wave. Sometimes they participate, sometimes with half strength, sometimes not at all. A "case" counts through the number of pixels in the surveillance zone in a frame. Therefore, you are sure to get a larger pixel area if there is a shimmering phase going on in that particular frame. However, if you are between two shimmering waves in time or outside the area of the actual shimmering at the nest, you are measuring the "quiescent", the "basic", actual a “non-shimmering” activity at that location and time. In any case, this is congruent with the background activity, which in turn also has a thickness in the motion profiles which can be characterized by envelope curves. This shows that in spectra we indicate probabilities with which a certain motion activity is depicted in the respective frame in the corresponding topographical nest area. If it is higher, it can be identified as locomotion or shimmering; if it is rhythmic, then it is shimmering. Lower motion activities in the spectrum under Pre- and PostMFA are always related to the basic activity, i.e. to a (at least momentary) quiescence.

Reviewer #1:

“but may even have been inhibited by the external threat scenario at work during this period.” How do you reach this conclusion?”

And

“some cohorts are frozen…”. Where am I supposed to see that? The red curves don’t seem to go much below the blue ones, and they wiggle up and down during the MFA.”

My comment: 

This conclusion of mine simply follows from the probability of occurrence with lower motion activity in individual frames during an experimental Pd session. The left-sided skewness (qs<1) under Pd conditions means that the lower envelopes of the corresponding motion time profiles are pulled down. Such points only occur in Pd (red in Fig 4), but not in nPd (blue) sessions. They are therefore below the extrapolated lower envelope of the blue point clouds. The only possible explanation for this is that there are several individuals, if not cohorts, that are even lowered in their motion activity by the presentation of the wasp dummy (because it only happens under this stimulus condition), i.e. by the external threat. Lower movement activity here means that the activity under Pd conditions is lower than the corresponding reference baseline activity under nPd. This is the reason that under Pd conditions we get a left-sided skewness value (qs<1), which in the context of MFA means that the bees at the surface of the nest are able to perceive external threatening objects in MFA mode. Again, similar to the "quiescence" explanation above, these are probability plots from the entire surveillance zone at a given time. The chapter on “lower motion activity” of the previous manuscript, which would have explained these phenomena more broadly, is omitted here in the updated manuscript along with the corresponding graph.

Reviewer #1:

“Through all this circumvoluted analysis of the data, what I take from this paper can be summarized quite simply:

-there may be a tremor that signals the start of the MFA

-the MFA spreads forth from and back to the mouth zone

-Bees don’t shimmer during the MFA (thus trading off between defence and other nest functions), but might instead freeze in response to threats

-Loss of shimmering is progressive but its return is abrupt, and occurs before the bees have fully settled back”

My comment: 

• I will thank Reviewer #1 for checking this paper in such an excellent way. This is also a competent summary of most of the outcomes in the paper. I hope that the revised manuscript here will also be much more transparent and easier to read in this sense.

• The paper now essentially describes two phenomena that belong very closely together: (A) evidence for the mass-flight-suspend-defensiveness hypothesis as an essential part, which also leads to the additional question of whether or not the perception of external threat is suspended during the mass flight phase. The paper also provides (B) evidence for a trade-off effect between MFA and defence preparedness in giant honeybees Apis dorsata, which describes the delicate decision-making situation of a colony to enter into MFA status.

• Both aspects (A,B) are so closely linked that it makes sense to address both in a single paper. In principle, this meant for me that only those measurement results from the studies were included in the paper that were also necessary for the presentation of the main topic, namely to what extent the empirical data of defensiveness and MFA correspond with the theoretical concept of a trade-off. I hope that the reviewers also agree with me that with even more cuts, the phenomenon of the main topic can no longer be explained comprehensibly with the text of the paper.

Reviewer #1:

“These conclusions are interesting and well-supported by the data, but the sample size is very small (3 videos analyzed, all from the same nest). How were these videos selected out of the ~80 that had apparently been recorded? Why use only a single nest?”

My comments:

• My selection was simple: There were only three episodes with a complete programme at preMFA, MFA and postMFA states. All other videos brought only partial time documents, but were certainly quite important to gain knowledge and experience to develop this kind of experimental approach with the presentation of dummy wasps for checking the defensiveness of the colony even during MFA. By the way, it is quite hard to gain such documents. To explain this, I have now added this text in the Method: “For these three selected episodes only, we were able to partially or fully apply a defence readiness check programme (see below), and were also able to record the entire MFA episode, including a lead-in and a lead-out of 10 minutes each.“ 

• The data volume is, indeed, very extensive and allows analysis of the data in the resolution of a second. The data refer to three episodes with more than 60 000 images each and refer to 50 Hz at a fairly high image resolution. With the help of the differential images, for example, it is quite possible to determine whether a single spread wing of a bee is moving actively or passively. For each of the five monitoring zones, more than 300 k of data were analysed. Furthermore, in addition to the motion time profiles, motion spectra were developed in two ways; essentially, the histogram-based spectra were used in this paper (with the exception of the trade-off analysis, where the results of the quantile-based spectra were also used for comparison, but these data are only provided for the survey in the table in the Supplementary material).

• The volume of the data sets used for three selected episodes (of one and the same experimental nest) shows that the evaluation of the data was time and resource consuming. Also, because the compilation of such observations with the experimental condition of a presentation of dummies was extraordinarily time-consuming and complex, the results also refer to only one nest and therefore represent what it is, a case study. 

• From our experience we can say with certainty that this case study is representative of other nests throughout the range of Apis dorsata. We could only use one nest for a detailed behavioural analysis for this type of study. However, it is also not to be expected that the present measuring method would have led to fundamentally different results at other nests.

• Such elaborate repeated measurements at other nests (also with these very detailed data analyses) could not have been carried out logistically and technically in the time frame available for an expedition of (only) several weeks in Nepal. Of course, the many observations from mass flights that we have collected over the years also provide the basic and expected insight that this collective behaviour of MFA is at least indistinguishable between populations in Nepal or Assam (India). 

• Incidentally, all my work on giant honeybees over the last 20 years, which has also been mainly concerned with analyses of collective behaviour, has been precisely such case studies, each with a number of n=1 (nests) studied. Such behaviours as MFA, gathering behaviour, or defensive behaviour such as shimmering behaviour do not have nest-specific properties.

Completing remarks for Reviewer #1:

• However, he mentioned the problem of repetition. I understand such a criticism and have had to address it every time in my behavioural analyses over the last twenty years. One must know that it is already a major undertaking, both technically and logistically, to provide three different episodes (of the same nest), especially because the present quantitative study goes into great detail, including of the experimental nest used. The experimental setup in front of the nest requires extraordinary effort, time and organisational luck that we were allowed to be there without disturbing the people in the immediate vicinity. To study multiple nests with the intention of analysing the inter-individuality of behaviours in a time window of 4 weeks in the wild on such a scale is practically impossible. In my opinion, it would also not yield any significant gain in knowledge. Of course, it can be said that every nest is different, and rightly so, because every nest is different simply because of its micro-ecology of the environment. This aspect undoubtedly makes this paper a ‘case study,’ as do many of the papers I have already published that analyse individual and collective behaviours in detail. 

• The problem is that the significance of the results relates to a large number of individuals and the statistical evidence relates to the inter-individuality of individuals rather than the inter-individuality of colonies. In my view, it is also sufficient for such a behavioural analysis to meet the topic of the study, because one can easily assume that the variability of the behaviour of the individuals does not differ from that of other nests in a uniform area such as Chitwan (at least probably not with regard to the behavioural parameters we studied). 

• Even though the present paper is such a case study, like all my previous publications with single nests, it is quite suitable to shed light on the traits we studied in mass flight behaviour in giant honeybees as a whole (i.e. in Apis dorsata and even in Apis laboriosa).

• While it is quite easy to photograph or even film mass flight behaviour from a greater distance (I also pointed out in the paper that we have probably seen many dozens of such mass flight events over time, also photographed and filmed), a continuous observation such as we have made, with a resolution of 4k and very close to a nest, where we can distinguish each individual bee on the surface, is a particular logistical challenge. Namely, it provides a continuous record of a mass flight (before, during and after). One of the greatest difficulties is to organise the appropriate mounts for the cameras and stimulation for such close-up recordings. Giant honeybees build their nests in a way that is extremely unfriendly to experimentation! 

• Incidentally, the hypothetical surmise of a trade-off between defensive behaviour and periodic nest rebuilding in a mass flight mode is a matter that sounds trivial and, at its core, plausible. But to prove this hypothesis with its hyperbolic basic function quantitatively (at least for a single nest) requires some technical considerations and quite sophisticated know-how to find and use the appropriate, adequate parameters. For this data analysis, we organised a very sound, broad volume ourselves; for each of the three episodes mfa1-3, at least 300 thousands of data were processed, so a total of almost 4 terabytes came together. 

Reviewer #2: 

“It is known that more than half A. dorsata colonies could perform daily periodic mass flights ranging in number from zero to six. This behaviour apparently play a role in cleansing and orientation flights of young worker bees”

My comment:

I don't understand the first sentence. The fact is, periodic mass flight is produced by virtually all Apis dorsata (and laboriosa) colonies, solely when they come into the need to improve their functional nest architecture. Involved are always younger bees from inside the nest and those from the surface layers of the bee curtain..

Reviewer #2: 

However, the role of MFA in nesting biology, behaviour as well as its ecological significance remains unclear. Therefore it is important to learn more about this phenomenon. This study investigate how a colony in MFA be mode while reconfigure its internal organization which can maintain its defense capability.” 

My comment:

The MFA behaviour as well as its ecological significance remains clear. It is quite well I think that the behaviour of MFA and its ecological importance is actually quite clear. It is quite well researched, even in homology with honey bees in general. I have included a few lines in the discussion anyway to summarise this for the reader. The main point of the present paper builds on these findings and asks a much more differentiated question, namely whether or not the perception of external threats is switched off during MFA. This is extremely important for those honey bees that build nests outdoors. The point is that there must be sophisticated behaviours here that counteract the otherwise high specialisation of nest predators. Because nesting in the open is damned dangerous, a la longue for the survival of the species as a whole. The present study therefore questions the possibilities that reed honey bees have perceived in the microdesign of their collective behaviour on the nest in the course of evolution in order to effectively mitigate the danger posed by highly specialised predators.

Reviewer #2:

“The authors using a dummy wasp to elicit shimmering behavior on the nest surface to measure whether external threats can be perceived collectively in MFA mode. From my personal perspective, this method is not advisable, I believe a dummy wasp visually could induce a wavy shimmering pattern, but there are many defensive ways in honey bees, for example, bee alarm pheromones and other physical alarm signals may be involved, honey bees detect predators also dependent on chemical signals from the predator at the same time. So when a colony in MFA mode, the flying bees might ignore visual stimuli, but it is possible that they increase their chemical defence, so it doesn't mean that the collective defense capability of the colony is undermined. On the contrary, such a state of social arousal is triggered preferably by external visual stimuli, usually leads to a synchronized, collectively well-organized arousal of many bees at the nest surface and even drives the wasps out of the area of the bee nest.”

My comment: 

• The main criticism of reviewer 2 is that the work did not take into account possible chemical signals that might be encountered by nest predators at the nest.

• At this point I have to say that the criticism of reviewer 2 does not apply to this study at all. There are three main reasons for this:

• Firstly, shimmering is a long-distance reaction that occurs, for example, towards honey buzzards when they fly past even at a distance of 30 meor more. Towards free-flying wasps (this is known from my work from Kastberger & Schmelzer 2008 onwards), shimmering is triggered when the wasp is at least one meter away from the nest in its natural scanning behaviour. If the wasp comes closer, the shimmering waves would become more intense in recruitment, but also in the repetition frequency of the waves. When the wasp hits the nest surface, it is caught by some bees and drawn into the bee curtain, where it dies from lack of oxygen and ball-heating by a considerable number of bees.

• Secondly, in the case of giant honeybees, any scent of a potentially predatory wasp that might emanate from the wasp e.g. at a distance of 20 to 100 cm from the nest could never reach the bees in the reaction time in which the shimmering is triggered (this latency is only hundreds of milliseconds between the sight of the wasp and the onset of shimmering); not to mention the honey buzzards in more than 10 m distance.

• However, it is more likely that the bees themselves are pheromonally active in shimmering via Nasonoff scent release, for this please read our paper Kastberger et al . Evidence of Nasonov scenting in colony defence of the Giant honeybee Apis dorsata Ethology 1998).

• The third point concerns the issue of the study itself: The dummies as applied in the experimental setup were specifically designed to elicit shimmering waves. And this is exactly what the presented work is about: a method is used to test and quantify the defensive readiness of the colony. On the one hand, a standardised stimulus is used to trigger defensive reactions, and on the other hand, these reactions are measured using selected motion parameters. Because there are several causes and graduations for motions in connection with the nest topography, the MFA and the defensive reaction of the colony, it is possible to pick apart any interferences by means of motion profiles and motion spectra. And precisely because the dummy was regularly presented (alternating with non-presentation) before, during and after mass flight throughout the observation period, it was possible to test the colony's changed defensive readiness as a result of the mass flight mode.

Reviewer #2: 

“So when a colony in MFA mode, the flying bees might ignore visual stimuli, but it is possible that they increase their chemical defence, so it doesn't mean that the collective defense capability of the colony is undermined.” 

My comment: 

Sorry, this is an assertion without any factual relevance, here the terms "chemical defence" and its "enhancement" are not defined. Otherwise, the present work is not concerned with the perception of flying bees during MFA, but with those that remained on the nest surface or returned from defecation or orientation flights.

One of another possible counter-arguments to this, as I may have understood this comment by Reviewer 2, would perhaps be that the swarming of many bees exudes danger to some addressees, so that in many cases this could trigger avoidance behaviour in predators. However, in the context of giant honey bees, there are also nest predators that specialise precisely in actively approaching swarms to strike prey. These are e.g. bee-eaters (Kastberger & Sharma 2000) or bats. However, these bats are so specialised that they are interested in the MFA at dusk, when swarms of stingless drones are encountered at certain times of the year.

Reviewer #2: 

“On the contrary, such a state of social arousal is triggered preferably by external visual stimuli, usually leads to a synchronized, collectively well-organized arousal of many bees at the nest surface and even drives the wasps out of the area of the bee nest.”

My comment: 

• Indeed, we have studied this aspect of shimmering extensively (Kastberger et al. Giant honeybees (Apis dorsata) mob wasps away from the nest by targeted visual patterns, Naturwissenschaften 2014). 

• However, this also has no relevance to the main topic in this paper. The main point here, as mentioned above, is that we used a standardised method of triggering defensive behaviour to measure the level of defensiveness in the colony. Artificial stimulation of shimmering and thus other subliminal reactions in the area of the bees' motions on the nest surface proved to be extremely suitable for this purpose, because here it was possible to make the defensiveness measurable under the various conditions before, during and after mass flight states of the experimental colony.

Thank you again and I hope I have clarified all questions and criticisms.

Kind regards

Gerald Kastberger

---

## [Decision Letter · Decision Letter 1]

1 Dec 2023

PONE-D-23-15840R1Giant honeybees (Apis dorsata) trade off defensiveness against periodic mass flight activityPLOS ONE

Dear Dr. Kastberger,

Thank you for submitting your manuscript to PLOS ONE. After careful consideration, we feel that it has merit but does not fully meet PLOS ONE’s publication criteria as it currently stands. Therefore, we invite you to submit a revised version of the manuscript that addresses the points raised during the review process. I made this decision to give you the opportunity to respond to the reviewer's few minor comments. Once this is done, I can accept the manuscript without involving reviewers again.

We look forward to receiving your revised manuscript.

Kind regards,

Wolfgang Blenau

Academic Editor

PLOS ONE

Journal Requirements:

Reviewers' comments:

Reviewer's Responses to Questions

**Comments to the Author**

1. If the authors have adequately addressed your comments raised in a previous round of review and you feel that this manuscript is now acceptable for publication, you may indicate that here to bypass the “Comments to the Author” section, enter your conflict of interest statement in the “Confidential to Editor” section, and submit your "Accept" recommendation.

Reviewer #1: (No Response)

2. Is the manuscript technically sound, and do the data support the conclusions?

Reviewer #1: Yes

3. Has the statistical analysis been performed appropriately and rigorously? 

Reviewer #1: Yes

4. Have the authors made all data underlying the findings in their manuscript fully available?

Reviewer #1: Yes

5. Is the manuscript presented in an intelligible fashion and written in standard English?

Reviewer #1: Yes

6. Review Comments to the Author

Reviewer #1: I thank the authors for their answers and revisions. The quality of the paper has increased immensely, it’s much more concise and a lot easier to read. I think the abstract could contain even less details about the methods used to reach the conclusions, but that may be a personal preference.

Below are a couple of details and typos that I noted:

L53: “In consequence the MFA intensity in mfa3 was significantly lower than in mfa1,2 which is an indication that the colony perceived external threats during MFA.” This sentence threw me off, I think because of the reversed relationship – I would rather argue that the MFA intensity is higher in mfa1,2 (than the baseline represented by mfa3).

L688-692 are repeated word for word L709-712. I think the 1st repeat was forgotten there?

L167: the lodge has a different name in the legend of Fig 1

L192: the “yellow” arrows are red.

L202-211: I’m not sure this section is necessary, and it seems to contradict previous statements (colonies need to be observed for 30min pre and post MFA according to this section, vs 10 min in the data presented).

L307: even … even

L473: <1Hz rather than 0.1, if I believe the x-axis of Fig 6?

L493-497: sentence repeated

L646-647: This sentence is weird, the grammar is wrong.

L652: bivouacs rather than bivaques

L658: the surface layer of the bee curtain

L704: even on the other side

L819: neither would be capable of shimmering. There and in other places, it is implied that the only defensive response available to the colony is shimmering. I think that is a bit of an over-generalization. If some bees can still detect the threat as suggested, they might still be able to individually fly out and sting/ball/harass the intruder, even if they have trouble building a strong collective response.

L842: first rather than lastly

L849: to an extent + to a certain degree

Jump from Fig 4 to Fig 6, Fig 5 not used?

Fig 7: red curves appear as dotted blue-red, making them hard to distinguish from the blue ones (and artboard visible in the back)

Just an idea - I feel like there could be a summary figure showing how both the MFA and defensiveness progress spatially across the nest (I mean a schematic representation, not another data-loaded figure) that would nicely wrap up the argument.

7. PLOS authors have the option to publish the peer review history of their article (what does this mean?). If published, this will include your full peer review and any attached files.

Reviewer #1: No

---

## [Author Response · Author response to Decision Letter 1]

6 Dec 2023

Rebuttal letter

I would like to thank reviewer #1 for his close review of the revised manuscript, for his comments and queries, which were very helpful in significantly improving the manuscript during the final revision of the paper.

Gerald Kastberger

Reviewer #1: I thank the authors for their answers and revisions. The quality of the paper has increased immensely, it’s much more concise and a lot easier to read. I think the abstract could contain even less details about the methods used to reach the conclusions, but that may be a personal preference.

Below are a couple of details and typos that I noted:

L53: “In consequence the MFA intensity in mfa3 was significantly lower than in mfa1,2 which is an indication that the colony perceived external threats during MFA.” This sentence threw me off, I think because of the reversed relationship – I would rather argue that the MFA intensity is higher in mfa1,2 (than the baseline represented by mfa3).

L688-692 are repeated word for word L709-712. I think the 1st repeat was forgotten there?

L167: the lodge has a different name in the legend of Fig 1

L192: the “yellow” arrows are red.

L202-211: I’m not sure this section is necessary, and it seems to contradict previous statements (colonies need to be observed for 30min pre and post MFA according to this section, vs 10 min in the data presented).

L307: even … even

L473: <1Hz rather than 0.1, if I believe the x-axis of Fig 6?

L493-497: sentence repeated

L646-647: This sentence is weird, the grammar is wrong.

L652: bivouacs rather than bivaques

L658: the surface layer of the bee curtain

L704: even on the other side

L819: neither would be capable of shimmering. There and in other places, it is implied that the only defensive response available to the colony is shimmering. I think that is a bit of an over-generalization. If some bees can still detect the threat as suggested, they might still be able to individually fly out and sting/ball/harass the intruder, even if they have trouble building a strong collective response.

L842: first rather than lastly

L849: to an extent + to a certain degree

Jump from Fig 4 to Fig 6, Fig 5 not used?

Fig 7: red curves appear as dotted blue-red, making them hard to distinguish from the blue ones (and artboard visible in the back)

Just an idea - I feel like there could be a summary figure showing how both the MFA and defensiveness progress spatially across the nest (I mean a schematic representation, not another data-loaded figure) that would nicely wrap up the argument.

Query 1 

L53: “In consequence the MFA intensity in mfa3 was significantly lower than in mfa1,2 which is an indication that the colony perceived external threats during MFA.” This sentence threw me off, I think because of the reversed relationship – I would rather argue that the MFA intensity is higher in mfa1,2 (than the baseline represented by mfa3).

My answer:

No, this assumption is not correct. Figure 4 clearly shows that the blue dot plots of panels C are lower than those of panels A and B during MFA. This is a very basic result and therefore I explicitly mentioned it in the Abstract. Also note that the ordinates are the same for all three MFA episodes and refer to the original data without any normalisation, which are the logarithmised values of the number of moving active pixels per monitoring zone.

Nevertheless, for reasons of easier readiness we edited instead…

“In consequence the MFA intensity in mfa3 was significantly lower than in mfa1,2 which is an indication that the colony perceived external threats during MFA.” 

.. the optional new version:

“In contrast to mfa1,2, in mfa3 the experimental regime started with the cyclic presentation of the dummy wasp only after the MFA had subsided. As a result, the MFA intensity in mfa3 was significantly lower than in mfa1,2, suggesting that a colony is able to perceive external threats during the MFA.”

Query 2 Repetition

Reviewer 1: L688-692 are repeated word for word L709-712. I think the 1st repeat was forgotten there?

My comment:

Thank you for pointing this out. I have restructured both parts to avoid repetition.

Instead ...(Ad Questions to MFA on timing and restructuring of functional nest architecture

“However, this also raises the question of whether a colony that is unable to develop defensive behavior is literally blind to external stimuli. Methodologically, this was investigated by exposing the experimental colony to a recurrent sequence of sessions (Fig 4) in which stimulation conditions in the form of presentation of a dummy wasp (Pd) alternated with periods of no external threat (nPd). Size, proximity, and speed of this dummy (Fig 1) were adjusted to produce shimmering behavior under semi-quiescent conditions [36, 39, 41]. This strategy of experimental arousal was used here to determine and comparatively measure the willingness or ability of cohorts at the nest surface to develop defensive responses under conditions of sQu and MFA.“

I apply …

“However, this also raises the question of whether a colony that is unable to develop defensive behaviour is literally blind to external stimuli. Methodologically, this was investigated by exposing the experimental colony to a cyclically recurring sequence of external threats in the form of the presentation of a dummy wasp (Pd) (Fig. 4).”

And instead...(Ad The three (a-c) recurrent excitation states of the giant honeybee nest under natural and experimental conditions:)

“In a nest, the semi-quiescence state (a) prevails most of the day, which is more or less the "normal", undisturbed state of life of giant honey bee colonies (S1-S4 movies). In this state, except for the mouth zone, the bees hang in the bee curtain [16, 19, 20, 23] in several layers with the head up and the abdomen down. By external, visual or even mechanical stimuli, the state of shimmering is triggered on demand [9, 27, 36, 39, 40], which is manifested by a wavelike sequence of motion salvos on the nest surface, each involving hundreds of bees simultaneously or sequentially. To an outside observer, such undulating patterns appear like a visible alarm signal that has evolved primarily to deter predatory wasps [9, 40] that would otherwise approach nests unrestricted to strike prey. Furthermore, there is evidence [26, 27] that with this behaviour, the colony is able to inform the nest mates about the current threat situation, even on both sides of the central honeycomb.”

“(b) This state of semi-quiescence of the nest and its readiness for defensive responses such as shimmering is interrupted at regular intervals during normal daytime operation [42-45] by the state of the MFA.”

(c) Finally, for the present study, the experimental colony was additionally exposed to a recurrent sequence of external stimuli during the sQu and MFA phases, which were themselves autonomous and self-organised [16, 84, 85], alternating stimulation conditions in the form of the presentation of a dummy wasp (Pd) with periods of no external threat (nPd). The size, proximity, and speed of this dummy (Fig 1) were adjusted to elicit shimmering behavior under sQu conditions [14, 23, 36, 39]. This type of external experimental stimulation was used here to comparatively measure the ability of cohorts at the nest surface to develop defensive responses also under MFA conditions.“

We apply…

„This state of semi-quiescence of the nest and its readiness for defensive responses such as shimmering is interrupted at regular intervals during normal daytime operation [42-45] by the state of the MFA (b). 

For the present study, the experimental colony was additionally exposed to a recurrent sequence of external stimuli (c) during the sQu and MFA phases, which were themselves autonomous and self-organised [16, 84, 85], alternating stimulation conditions in the form of the presentation of a dummy wasp (Pd) with periods of no external threat (nPd). The size, proximity, and speed of this dummy (Fig 1) were adjusted to elicit shimmering behavior under sQu conditions [14, 23, 36, 39]. This type of external experimental stimulation was used here to comparatively measure the ability of cohorts at the nest surface to develop defensive responses also under MFA conditions. “

Query 3

L167: the lodge has a different name in the legend of Fig 1

My comment:

Eden Jungle Resort is nowadays the correct name

Done

Query 4

L192: the “yellow” arrows are red.

My comment:

Fig 3 was additionally edited!

Edited text:

(Fig 1B-C; see the red arrows in Fig 2A2a, and in Fig 3).

Done

Query 5

Reviewer 1: L202-211: I’m not sure this section is necessary, and it seems to contradict previous statements (colonies need to be observed for 30min pre and post MFA according to this section, vs 10 min in the data presented).

My comment:

Instead…

„In this paper we examine the question of how the collective defense changes during MFA of an Apis dorsata colony. To do this, it is necessary to start observing the nest in semi-quiescence before the MFA, preferably for at least half an hour, before the massive upheaval in the nest takes place and new functions and positions are assigned to the individual bees. But this then also presupposes that one can realistically estimate the beginning of such a process, which requires some experience with giant honeybees. In the field, we succeeded in predicting the start of such a MFA with a span of one hour, taking into account comparable environmental and the general weather conditions and, in particular, the ambient temperature. Likewise, for an in-depth analysis, it is useful to observe the behavior of the experimental colony for at least one hour also after the MFA has ended.“

I apply …

“In this paper we investigate the question of how collective defence changes during the MFA of an Apis dorsata colony. To do this, it is necessary to start observing the nest in semi-quiescence before the MFA, preferably for at least half an hour, before the massive upheaval in the nest takes place and the individual bees are assigned new functions and positions. However, this also requires some experience with giant honey bees to be able to realistically estimate the start of such a process. In the field, we have been able to predict the onset of such an MFA with a span of one hour, taking into account comparable environmental and general weather conditions and especially the ambient temperature. It is also useful for a detailed analysis to observe the behaviour of the experimental colony for at least one hour after the end of the MFA. However, for practical reasons, the actual recording times were somewhat shorter than the actual observation time we spent at the nest.”

My comment:

I do not want to omit this chapter as it underlines the enormous effort, work and patience required to correctly interpret, explain and understand the behaviours associated with MFA. I have therefore also made some minor changes to eliminate the apparent contradictions with earlier statements about recording times.

Query 6

Reviewer 1: L307: even … even

My comment:

Instead…

“(sometimes the giant honeybees even collect nectar and pollen even at night time [60, 61]).”

I apply…

“(sometimes the giant honeybees collect nectar and pollen even at night time [60, 61]).”

Query 7

Reviewer 1: L473: <1Hz rather than 0.1, if I believe the x-axis of Fig 6?

My comment:

Instead…

“In Fig 6A,B this motion turbulence was identified by its frequency components (Eq#4e) in the range <0.1 Hz, and was found to be distributed in a specific topographic pattern over the nest surface.”

I apply…

“In Fig. 6A,B, this motion turbulence was identified by its frequency components (Eq#4e) in the range from 100 Hz down to 0.1 Hz and was found to be distributed in a specific topographic pattern over the nest surface.”

My comment:

The diagrams concentrate on logarithmic units between 0.1 and 100 Hz, the calculations of course continue on both sides.

Done

Query 8

Reviewer 1: L493-497: sentence repeated

My comment:

Instead...

“In the present work, histogram-based movement spectra are used. This allows a more detailed description of how the nest evolves from the state of sQu to the state of MFA in the course of the experimental sessions, and how the MFA intensifies to its peak and then returns to sQu. As shown for episode mfa1 in Fig 7A, the motion spectra of the nPd phases (blue lines) were monomodal and even Gaussian when the size classes were scaled logarithmically. As shown for episode mfa1 in Fig. 7A, the motion spectra of the nPd phases (blue lines) were monomodal and even Gaussian when the size classes were scaled logarithmically.”

I apply..

“In the present work, histogram-based movement spectra are used. This allows a more detailed description of how the nest evolves from the state of sQu to the state of MFA in the course of the experimental sessions, and how the MFA intensifies to its peak and then returns to sQu. As shown for episode mfa1 in Fig 7A, the motion spectra of the nPd phases (blue lines) were monomodal and even Gaussian when the size classes were scaled logarithmically.”

Done

Query 9

Reviewer 1: L646-647: This sentence is weird, the grammar is wrong.

My comment:

I don't think that the grammar is wrong. However, I have corrected this obvious inconsistency by changing the wording slightly. I admit that the first sentence of a chapter should be a bit more general.

Instead…

“Mass flight in giant honeybees [35, 42-45, 47] manifests itself to the external observer that massive swarming behaviour takes place around such active nests, caused by triggers internal to the nest but not by any external threat.”

I apply…

“An attentive observer will not fail to notice that the swarming during the so-called mass flight of the giant honeybees [35, 42-45, 47] is caused by triggers within the nest and not by external threat.”

Query 10

Reviewer 1: L652: bivouacs rather than bivaques

Instead…

“In giant honeybees, absconding occurs regularly during migration, at comb-right nests or at bivaques [12, 13].”

I apply…

“In giant honeybees, absconding occurs regularly during migration, at comb-right nests or at bivouacs [12, 13].”

Done

Query 11

Reviewer 1: L658: the surface layer of the bee curtain

My comment:

I agree that just to avoid repetition it is fine to make this edit

Instead…

“In giant honeybees nests, during MFA state, bees slip through the mesh of the bee curtain from the inside to the outside (see S1 Movie and description at 0.000 min), crawl around on the nest surface, fly off to circle the nest in a swarm for at least half a minute, and finally land back on the nest surface.”

I apply…

“In giant honeybees nests, during MFA state, bees slip through the mesh of the bee curtain from the inside to the outside (see S1 Movie and description at 0.000 min), crawl around on the nest surface, fly off to circle the nest in a swarm for at least half a minute, and finally land back on the surface layer of the bee curtain.”

Query 12

Reviewer 1: L704: even on the other side

Instead...

„Furthermore, there is evidence [26, 27] that with this behaviour, the colony is able to inform the nest mates about the current threat situation, even on both sides of the central honeycomb.” 

I applied..

“Furthermore, there is evidence [26, 27] that with this behaviour, the colony is able to inform the nest mates about the current threat situation, even on the other side of the central honeycomb.” 

Query 13

Reviewer 1: L819: neither would be capable of shimmering. There and in other places, it is implied that the only defensive response available to the colony is shimmering. I think that is a bit of an over-generalization. If some bees can still detect the threat as suggested, they might still be able to individually fly out and sting/ball/harass the intruder, even if they have trouble building a strong collective response.

“In either case, this would mean that both cohorts remain susceptible to external threats even in MFA mode, but behave in opposite ways. Neither, however, would be capable of a defensive response.” 

My comment:

I want to leave this passage as it is. It is correct to say this in the context of MFA: “Neither, however, would be capable of a defensive response”. 

We have conducted many experiments on this topic (but have not yet published them). We found that shimmering is always the first option for defence during the day, as long as the ability to shimmer has been developed since early morning and as long as the stimulation or threat intensity remains low.

When the intensity of the threat increases, for example when the visual threats in terms of geometric size and angular velocity increase, it takes a few tens of seconds to recruit a crowd of bees to fly against these visual patterns in front of the nest. When the threat level increases so immensely, this is a completely different category than when "just shimmering" is triggered, and also requires a much higher level of defence. The phenomenon of individual bees flying out of the nest to attack is obviously rare or non-existent.

Of course, there are always individual bees that catch and sting you when you are close to the nest or even further away. We experienced this not only with Apis dorsata, but also with Apis laboriosa, e.g. in Chaku, Nepal itself at a distance of 300 metres from the nest, and the great Himalayan river Bhotekoshi was even between us. However, these are always members of a small swarm that are released and search for their targets, then also singularized.

Query 14

Reviewer 1: L842: first rather than lastly

My comment:

I would like to introduce a different idea here. I have restructured the wording accordingly.

Instead…

“On the other hand, the ability to defend itself is in Giant honeybee nests completely suspended for a some minutes. Defensiveness lastly starts with all nest members, or at least with a strategically selected cohort, that keeps their eyes open to perceive threats coming from outside.”

I apply…

“On the other hand, the ability to defend itself is in Giant honeybee nests completely suspended for a some minutes. Finally, defence readiness begins when all nest members are ready and able to do so again, or at least with a strategically selected group that keeps its eyes open to detect threats from outside.”

Query 15

Reviewer 1: L849: to an extent + to a certain degree

Instead..

“Interestingly, the mouth zone is a kind of exception here (Fig 10A4,B4), less concerned with handling the trade-off between MFA and defensiveness. It retains to some extent its function to a certain degree as a hub for various important tasks in the nest, such as foraging, guarding, but also in this respect to trigger the signal for the beginning and end of mass flight.”

I apply…

“Interestingly, the mouth zone is a kind of exception here (Fig 10A4,B4), less concerned with handling the trade-off between MFA and defensiveness. To a certain extent, it retains its function as a hub for various important tasks in the nest, such as foraging, guarding, but also in this respect to signal the start and end of the mass flight.”

Done

Query 16

Reviewer 1: Jump from Fig 4 to Fig 6, Fig 5 not used?

My comment

In 462 (Results) Fig. 5 has already been mentioned

“Although the motion amplitudes were normalized separately in each of the episodes (Fig 5),…”

In fact, I have omitted to cite this Fig 5 several times. I have now made up for this (in Methods, Results and Discussion). Now I added it also in the Methods (new line 301) to have a continuation in the sequence of Figure citation; in Results (line 460), in Discussion line 715,783,791, 794 e.g. “(Figs 4AB,5AB in contrast to Figs 4C,5C)”

Query 17

Reviewer 1: Fig 7: red curves appear as dotted blue-red, making them hard to distinguish from the blue ones (and artboard visible in the back)

My comment

 I have widened the red line three times (to “3 mm”). The artboard is now no longer visible in the back.

Query 18

Reviewer 1: Just an idea - I feel like there could be a summary figure showing how both the MFA and defensiveness progress spatially across the nest (I mean a schematic representation, not another data-loaded figure) that would nicely wrap up the argument.

My comment:

Thank you for this proposal. I thought about this idea for a while, but I have to admit that any suggestion I had in mind would either have been too trivial or would even have led to the wrong results. Any generalisation I tried would not, in my opinion, summarise the results satisfactorily. I'm sorry about that. Unfortunately, we have to make do with the line plots and motion spectra (and films) offered in the paper.

Additionals:

I have changed some text details in the captions of the figures (such as of Fig 1, 3, 5). 

---

## [Editor Report · Decision Letter 2]

25 Jan 2024

Giant honeybees (Apis dorsata) trade off defensiveness against periodic mass flight activity

PONE-D-23-15840R2

Dear Dr. Kastberger,

We’re pleased to inform you that your manuscript has been judged scientifically suitable for publication and will be formally accepted for publication once it meets all outstanding technical requirements.

Kind regards,

Wolfgang Blenau

Academic Editor

PLOS ONE
---

## [Editor Report · Acceptance letter]

25 Mar 2024

PONE-D-23-15840R2 

PLOS ONE

Dear Dr. Kastberger, 

I'm pleased to inform you that your manuscript has been deemed suitable for publication in PLOS ONE. Congratulations! Your manuscript is now being handed over to our production team.

Kind regards, 

on behalf of

Dr. Wolfgang Blenau 

Academic Editor

PLOS ONE